# EvoDevo: Past and Future of Continuum and Process Plant Morphology

**Rolf Rutishauser**

Department of Systematic and Evolutionary Botany, University of Zurich, CH-8008 Zurich, Switzerland; rutishau@systbot.uzh.ch

**Abstract:** Plants and animals are both important for studies in evolutionary developmental biology (EvoDevo). Plant morphology as a valuable discipline of EvoDevo is set for a paradigm shift. Process thinking and the continuum approach in plant morphology allow us to perceive and interpret growing plants as combinations of developmental processes rather than as assemblages of structural units ("organs") such as roots, stems, leaves, and flowers. These dynamic philosophical perspectives were already favored by botanists and philosophers such as Agnes Arber (1879–1960) and Rolf Sattler (*1936). The acceptance of growing plants as dynamic continua inspires EvoDevo scientists such as developmental geneticists and evolutionary biologists to move towards a more holistic understanding of plants in time and space. This review will appeal to many young scientists in the plant development research fields. It covers a wide range of relevant publications from the past to present.

**Keywords:** process philosophy; scientific perspectivism; developmental genetics; plant structure ontology; homology; land plant phylogeny; morphological misfits; flower; phyllotaxis; *Utricularia*

---

## 1. Introduction

Plant morphology is the study of the physical form and structure of plants. Literally, the term "morphology" is derived from the Greek roots: *morphe*, which means form and/or structure, and *logos*, meaning discourse or investigation [1]. Plant morphology usually includes plant anatomy, which is the study of the internal structure of plants, as seen at the microscopic level. Plant morphology comprises both adult structures as well as their development. Two aspects of plant development can be distinguished: (i) the development of whole plants from embryos to fertile adults, known as ontogeny; and (ii) the development of each part, such as a foliage leaf or a flower that starts as a group of undifferentiated stem cells (called meristems or primordia). Unlike most multicellular animals, plants usually have an open bauplan with long-lasting apical growth, continued branching, and repetition of subunits (plant organs) such as leaves, stems, roots, and flowers. This is especially true for seed plants, whereas ferns (as another group of vascular plants) lack flowers and seeds (see Section 4.1).

This essay is written by a plant morphologist and historian of botany. It is a contribution to a special issue in the journal PHILOSOPHIES, edited by Alessandro Minelli on "Renegotiating Disciplinary Fields in the Life Sciences", with another three contributions [2–4]. The four PHILOSOPHIES papers demonstrate the need for scientific pluralism in biological disciplines such as evolutionary biology, developmental genetics, and taxonomy. They focus on recent and ongoing debates in biology (including philosophy of biology) that reveal shortcomings of traditional explanatory frameworks. The gene-centered perspective should be complemented by an organism-centered biology based on developmental processes. My contribution—focussing on plants—emphasizes that our knowledge of comparative biology, especially comparative plant morphology, is based on two centuries of knowledge and should not be neglected by the new generation of EvoDevo scientists.

The second part of this essay is a general introduction on process philosophy and continuum views in biology. It prepares the way of thinking (including scientific perspectivism) needed for a paradigm shift in plant morphology as a valuable sub-discipline of EvoDevo.

The third part is dedicated to Agnes Arber and Rolf Sattler as early proponents of continuum and process morphology in plants. Process thinking and continuum approaches have implications for molecular and developmental genetics. Instead of asking which genes control the development of each single plant part (such as a foliage leaf or a flower), we may concentrate on combinations of developmental processes that create the whole plant body with, for example, leaves and flowers as iterated products.

In the fourth part, various case studies from the plant kingdom will be presented illustrating the need for process thinking and continuum approaches. This part focusses on several new publications on vascular plants elucidating the regulation of plant architecture by genes. For example, the bladderworts (genus *Utricularia*) are a well-known group of carnivorous flowering plants that attract students because of their unique features in biology, morphology, and developmental genetics. More specifically, the aquatic *Utricularia gibba* and the terrestrial *U. reniformis* were chosen as model organisms by geneticists, allowing further insights into developmental and evolutionary processes that may be relevant for land plants in general.

## 2. Philosophy of Biology: Complementarity, Continuum, and Process Thinking in Development and Evolution

### 2.1. Philosophy of Biology

Is there any connection between philosophy *and* biology? This question is asked by Rolf Sattler (*1936), who has taught philosophy of biology for many years at McGill University (Montreal) and published a book on that subject [5]. Biological research is influenced by philosophical assumptions such as either/or logic, fuzzy logic, structure/process dualism, or its transcendence. Thus, there are interactions between philosophy and empirical findings. These interactions are the subject of what has been referred to as biophilosophy [5,6].

Sattler [7] writes: "It seems impossible to carry out any scientific investigation without philosophical assumptions. These assumptions are often taken for granted, and scientists may not even be aware of them. Therefore, it is one of the tasks of philosophy of science and biophilosophical investigations to make us aware of these assumptions that usually are part of a worldview. Unless one is aware of these assumptions and the associated worldview, one cannot evaluate them and perhaps change them if they appear inappropriate and no longer supported by empirical evidence."

Agnes Arber (1879–1960) was a British biophilosopher, historian of botany, and plant morphologist [8–10]. At the 16th International Botanical Congress in 1999, a symposium was held on the relationship between Arber's work and new explanatory models for plant development. The symposium contributions were published in *Annals of Botany* (see e.g., [11–15]). Bruce Kirchoff [11] (p. 1103), one of the organizers of the symposium, noted that new techniques in biology "have shifted our research focus toward molecular, physiological, and genetic aspects of plant development. What is forgotten in times like these is that our interpretation of the data is contingent on the models we bring to them. Data alone do not support or refute any theory. Agnes Arber's work is a powerful reminder of this fact. She constantly reminds of the relationship between data and theory by returning again and again to the data to view them in new ways, reinterpreting them in the light of new theories." The systems biologist Arthur Lander [16] stated more recently: "Models do not arise by logical inference from data; they are acts of human creation."

### 2.2. Scientific Perspectivism and Complementarity in Biology

"Individual scientists, and science as a whole in its specific historical context, only ever see a small proportion of all perceivable events or phenomena. Our understanding always remains

incomplete and biased by our personal history and societal context." Johannes Jaeger (2017, p. 139) [17]

Various philosophers and scientists accepted two or more complementary views, perspectives, or modes to describe and interpret form and function of living matter, including growth of plant structures [2,5,17–22].

Scientific perspectivism accepts different positions as perspectives on reality that complement one another. Although perspectivism is often used in colloquial speech, it is not common in natural science including biology [4,5,13,23]. Perspectivism accepts every insight into nature as one perspective (but not the only one) to see and explain biological phenomena. Different perspectives (also called approaches, models) complement each other, rather than compete with each other, although not all of them are meant to be equal approximations to what really occurs in nature. Perspectives may be even somewhat contradictory to each other. As long as they are supported by facts, they should be accepted as explanatory hypotheses, because they illuminate other facets of reality that may be useful and important as well.

For example, Minelli [24] and Pradeu [25], in a historical and evolutionary context, discussed the question: What is an individual in biology? Minelli's conclusion: "None of the concepts thus far advanced by biologists or philosophers of life covers in satisfactory way all instances and aspects of biological individuality". Thus, two or more complementary views may be justified for description of modular organisms, such as a coral or a branched woody plant: Each one may be viewed as a population of many individuals as well as one but highly complex individual [26,27].

Perspectivism was anticipated by Arber [28–30] and earlier philosophers who stressed the "coincidence of contraries". This term describes the somewhat astonishing situation that even biologists are allowed to label phenomena of living organisms (e.g., a leaflet of a plant) with seemingly contradictory terms. Perspectivism is closely related to as-if-ism, pluralism, conceptual nominalism, and Woodger's [31] map analogy.

Examples of complementary perspectives or world views as used in plant morphology will be presented in Section 3.3.

## 2.3. Clear-Cut Language Versus Fuzzy Concepts in Natural Sciences Especially Biology

In contrast to perspectivism, there is the view based on conceptual realism (essentialism), assuming that structural categories are immanent in life, forming crisp or distinct sets, i.e., terms with non-overlapping connotations and without intermediates. This school of thought (also called substance ontology or substance metaphysics) is common in biology and other natural sciences. Looking for essential "units" coincides with "either/or thinking" as expressed by Aristotle's Law of Contradiction ("A cannot be both A and not-A"), which is the basis of most ordinary discursive–logical reasoning [20,30,32,33]. According to this approach, natural sciences require a clear-cut language consisting of well-defined terms and notions that allow either/or decisions (as exemplified by the plant structure ontology, see Section 3.1).

No doubt, we need a consistent terminology in many disciplines of natural sciences in order to become understandable, even in plant morphology. However, drastic evolutionary changes in bauplans of living organisms may require fuzzy rather than clear-cut concepts of organ identity for description [4,13,34–37], calling for a fuzzy approach while searching for continua in time and space. This second approach coincides with process philosophy (also called process metaphysics) in cognitive psychology and systems biology. For an illustration, let us take two examples from colloquial speech: (i) The concept of biological individuality [24,25], which applies well to most human beings (except for Siamese twins), becomes fuzzy when used to describe colonial organisms, including corals and trees (and even social insects such as the honeybee). (ii) The behavioral biologist Bernhard Hassenstein (1922–2016) [23] pointed out that life must be seen as "injunction", i.e., as a concept that cannot be defined by a clear-cut set of properties. Illustrating the problem, Hassenstein asked the question "How many grains result in a heap?" There is no clear-cut answer to this question. It depends on our

perspective (including the size of the grains), if five, 20, or 50 grains are needed as a minimum to get a heap.

*2.4. From Process Philosophy to Process Thinking in Biology*

The tradition of substance ontology has dominated philosophers' views of reality so far. However, it seems that times may be changing. According to Sattler [7]: "A structure is not seen as having processes, a structure is seen as process(es)." Then, there is no longer a structure–process dualism. Process philosophy argues that processes are more fundamental than substance (i.e., static things or entities) [38,39]. Jaeger [17] (p. 138) gives a short introduction to process philosophy as part of systems biology: "In its most pragmatic form, process philosophy states that it is useful and important to study nature in processual terms. In other words, while it is important to know what a system is made of, it is the interactions between its components that define a system's behavior (its dynamical repertoire) and its potential for future change . . . Thus, things can only be studied as parts of processes, which is exactly what systems biology is supposed to do."

This process philosophical approach was used by Nicholson and Dupré [40] for all kind of organisms (even for viruses): "The living world is a world of process rather than a world of things." According to the process(ual) philosophy of biology, there are no misfits; everything fits, because everything can be understood in terms of developmental processes such as, for example, branching and articulation. Baedke and Mc Manus [41] are quite aware of this trend towards process thinking in biosciences when they write: "Recently, a growing number of philosophers of science, first and foremost John Dupré and colleagues [40,42,43], have emphasized the need to adopt a processual perspective for studying the complex dynamics and close connectedness of living systems."

Alfred North Whitehead (1861–1947) was one of the founders of modern "process philosophy" [38,39,44,45]. However, when searching for Whitehead's influence on well-known evolutionary biologists such as Ernst Mayr, one is disappointed. Mayr [46] (p. 38) wrote only one sentence about Whitehead, calling him "a peculiar mixture of a mathematician and a mystic" (in German "eine sonderbare Mischung von Mathematiker und Mystiker"). The science historian Ilse Jahn [47] in her superb "Geschichte der Biologie" did not mention either "process philosophy" or Whitehead's contributions at all.

There is a growing group of botanists who internalized process thinking in their research attitudes during the last two to four decades. Mabberley and Hay [48] (p. 122) wrote: " . . . it has become increasingly accepted that in plants, it is better to consider an organism and what are perceived by many as its component parts at any one time as a snapshot in a dynamic transformation of growth, maturity, and senescence." In botany, process thinking is known as process plant morphology or dynamic morphology [49] (see more in Section 3.6).

Johanna Seibt [33] admits: "While process philosophers insist that all within and about reality is continuously going on and coming out, they do not deny that there are temporally stable and reliably recurrent aspects of reality." In biosciences, there is a diversity of hierarchical descriptions with respect to structures (see e.g., plant structure ontology, as mentioned in Section 3.1.). Baedke and Mc Manus [41,50] have shown that developmental processes in organisms can also be viewed as part of time scale hierarchies (labelled as "dynamic hierarchies"), including processual hierarchies between genotype and phenotype. Somewhere between the structural approach (substance ontology) and process philosophy (process ontology) is the concept of dynamical patterning modules, as described by Benitez et al. [51]. These are defined as "sets of conserved gene products and molecular networks in conjunction with the physical morphogenetic and patterning processes they mobilize." Similar "developmental units" were proposed for modular organisms, including land plants. Minelli [4] called them "organizational modules"; Lacroix et al. [49], following the Russian paleobotanist S.V. Meyen, labelled them as "repeating polymorphic sets".

*2.5. Evolutionary Developmental Biology (EvoDevo)*

> "Organization is a continuum in the physical world. Organization is also a continuum in the ontogenesis and reproduction of the individual organism and in the phyletic line of which it is a component." C.W. Wardlaw [52] (p. 371)

**Evolution involves the transformation of ontogenies.** Sattler [7] argues as follows: "A problem of much evolutionary morphology is that, contrary to what is often said, structures are not directly derived from one another in a strict genealogical sense: organisms give rise to one another, but not structures … Although this seems obvious, it has often been ignored, but now it appears to be increasingly recognized in EvoDevo (evolutionary developmental biology): ontogenies change during evolution. Zimmermann [53] knew this long ago when he emphasized hologeny, which refers to the continuum of successive ontogenies. He saw clearly that evolution involves the transformation of ontogenies."

Within the last 30–40 years, evolutionary biology/morphology became known as EvoDevo by the arrival of developmental and molecular genetics. Minelli [4] writes: "One of the most sensational results was the discovery of the involvement of homologous genes in the development of such different organisms as mouse and fruit fly. The increasingly accessible contents of the black box between genotype and phenotype proved to be of utmost interest not only for development biologists, but also for evolutionary biologists". According to Minelli [4], the emergence of EvoDevo as a new interdisciplinary research field is conventionally fixed by the books of Raff and Kaufman [54] and Hall [55]. A concise introduction to EvoDevo is given by Minelli [56], pointing to the following four aspects [57–61]:

(i)　In the last 20 years since 2000, there has been a rapid growth of EvoDevo as a new approach to understanding the evolution and development of organismic form.
(ii)　To a considerable extent, EvoDevo deals with developmental genes, their evolution, and their expression.
(iii)　EvoDevo explains the arrival of the fittest, whereas Darwinism explains its survival.
(iv)　There is a strong need to focus on the phenotype, which is at the same time the product of development and the direct target of selection.

Ivan Amato [3] in his new PHILOSOPHIES essay tried to answer the following questions: "From an epistemological point of view, what is the relationship between EvoDevo and previous biological tradition? Is EvoDevo the carrier of a new message about how to conceive evolution and development?" It is worth reading Amato's review, because EvoDevo can be defined differently. While referring to new evolutionary theories in and beyond neo-darwinism, Amato presents 17 (seventeen!) different perspectives that complement each other to some degree, among them systems biology [17], epigenetic EvoDevo sensu Müller and Newman [62], and self-organization of biological complexity [63,64].

## 3. Classical, Continuum, and Process View in Plant Morphology

*3.1. Classical Plant Morphology as the Tradional Mainstream View*

It was not the very versatile Goethe (being poet as well as scientist), but mainly the German plant morphologist Wilhelm Troll (1897–1978) [65], who accepted [66] (p. 246): "that leaves, shoots, and the like are fundamental building blocks of the plant archetype: the starting point (in the ideal sense rather than the modern temporal/evolutionary sense) from which all plants are derived." Traditional botanists such as Wilhelm Troll were used to neatly distinguish foliage leaves (phyllomes), stems (caulomes), and roots as the three most obvious structural categories of vascular plants, as distinct and invariant modules that build up higher plants. Classical plant morphologists ("typologists") usually take for granted that all (or most) multicellular vascularized structures found in ferns and seed plants can be identified (homologized 1:1) with either a leaf, a stem, or a root [15,67,68].



The developmental geneticists Kinoshita and Tsukaya [69] mentioned the position criterion for the distinction of leaves and stems (shoots) in flowering plants and gymnosperms: "The aerial part of (typical) seed plants is called the shoot, which is composed of stems, leaves, and axillary buds. These are produced by indeterminate activity in the shoot apical meristem (SAM), whereas the morphogenesis of leaves depends on determinate activity of leaf meristems." Based on the publications by Kaplan [67,68], Dengler and Tsukaya [70], and Frangedakis et al. [71], the developmental geneticists Cruz et al. [72] (p. 1) summarized the tenets of classical plant morphology as follows: "Vascular plant organs are classically defined based on their position; on their tissue organization (symmetry axes and vascular tissue); and on the presence, position, and activity of their meristems. With these criteria, leaves are lateral determinate organs generally with an abaxial–adaxial asymmetry, and these features seem to generally apply well to leaves in seed plants. On the other hand, shoots are characterized by indeterminacy and are marked by the expression of *Class I KNOTTED-LIKE HOMEOBOX (KNOX)* genes in the shoot apical meristem (SAM)."—It is interesting to realize that the very last criterion to distinguish stems (shoots) and leaves mentioned by Cruz et al. [72] is provided by developmental genetics (see Section 3.5).

Mainly based on classical plant morphology is the **Plant Structure Ontology (PSO: www. plantontology.org),** which is thought to be a controlled vocabulary of botanical terms describing morphological and anatomical structures representing organ, tissue, and cell types and their hierarchical relationships [34,73]. It was developed by the Plant Ontology Consortium in response to the rapid proliferation of molecular sequences and databases, which has created data access problems for biologists. The main intent was to create a unified and hierarchical vocabulary of plant morphology and anatomy that can be used to describe spatial and temporal aspects of gene expression, because—traditionally—the same terms are sometimes applied to different plant structures in different taxonomic groups. Each of the nearly 60,000 structural terms is linked to a PO number. For illustration of the complexity of the PSO, let us start with the definition of a **whole plant** (PO: 0000003): "A plant structure (PO:0005679) that is a whole organism." Now, we need to know what a **plant structure** is. PSO provides the following definition (PO:0009011): "An anatomical structure that is or was part of a plant, or was derived from a part of a plant." Let us go on with the definition of **plant organ** (PO:009008) as provided by PSO: "A multi-tissue plant structure (PO:0025496) that is a functional unit, is a proper part of a whole plant (PO:0000003), and includes portions of plant tissue (PO:0009007) of at least two different types that derive from a common developmental path." Then let us continue with **cardinal organ part** (PO:0025001): "A cardinal part of multi-tissue plant structure (PO:0025498) that is a proper part of a plant organ (PO:0009008) and includes portions of plant tissue (PO:0009007) of at least two different types." Now we look for the PSO definitions of **leaf** as one of the three main plant organs in classical plant morphology: **Phyllome** (PO:0006001) and leaf (PO:0025034) are defined as follows: A phyllome is "a lateral plant organ (PO:0009008) produced by a shoot apical meristem (PO:0020148)". More specifically: A leaf is "a phyllome (PO:0006001) that is not associated with a reproductive structure". With respect to vascular plants, the leaf is specified as **vascular leaf** (PO:0009025), i.e., as leaf "having vascular tissue" with the following circumscription: "In angiosperms, commonly thought of as one of the three basic parts of the seed plant body, a structure usually of determinate growth, without secondary thickening, and of superficial origin, often flattened and photosynthetic in part, and in the axil of which is found a bud. Occurs in the sporophytic phase of a plant life cycle." Let us now look for subunits (e.g., leaflet) of leaves. As an example, the short PSO definition of **compound leaf** (PO:0020043) is "a leaf having two or more distinct leaflets that are evident as such from early in development." As a daughter term, the definition of **leaflet** (PO:0020049) is given: "A cardinal organ part (PO:0025001) that is one of the ultimate segments of a compound leaf (PO:0020043)."

In spite of the original aim of the PSO [73] to provide a "controlled vocabulary of botanical terms describing morphological and anatomical structures", the reader of the paragraph above may have been confused by the hierarchical complexity of structural terms used in botany. A critical review on possible shortcomings of PSO was already presented by Kirchoff et al. [34]. Besides the danger of

simplification and some weaknesses due to the hierarchical order of all the structural terms, the PSO does not cover alternative shoot concepts (e.g., phytomere) and bauplan oddities such as leaf-shoot intermediates and other kinds of morphological misfits (see below and Section 3.3).

*3.2. Morphological Misfits (Bauplan Oddities) as Test for Classical Plant Morphology*

Classical plant morphology as understood by Wilhelm Troll was based on essentialism, which is also known as conceptual realism [46]. According to David Baum [66]: "Essentialism implies that parts, a few more or less universal types, are the universal building blocks of vascular plants".

Schneider [74] presented the following criticism of a too dogmatic way of classical plant morphology: "It is important to keep the influence of classical morphology in mind, because it is still very influential especially as a result of the remarkable efforts of Don Kaplan [67,68], who provided access to this knowledge to a mainly English reading audience. However, classical morphology is not well aligned to the concept of Darwinian evolution. It is very important to keep in mind that the meaning of 'primitive' in a typological context is not synonymous with 'ancestral' in an evolutionary context".

Bauplans (also called body plans) are generalizations of our thinking and classifying mind. However, there is no doubt that certain animals and plants evolved structures (organs, appendages) that cannot be sensibly accommodated in traditional descriptions. Some plant groups were referred to as **morphological misfits** by Adrian Bell [75]. He pointed to the fact that morphological misfits are "misfits to a botanical discipline, not misfits for a successful existence". Morphological misfits are also observable in animals [36]. Various morphological misfits emerged as morphological key innovations (perhaps "hopeful monsters") that gave rise to new evolutionary lines of organisms [76–78]. The concept of "morphological misfits" is an eye-catcher that allows labelling of all kinds of morphological deviations in the wild, mainly based on major genetic changes such as homeosis (i.e., ectopic gene expression in a seemingly wrong position) and other kinds of developmental repatterning [36,79,80]. Morphological misfits are found in various aquatic vascular plants such as the duck-weeds (*Lemna* and allies in Araceae) [81] and the river-weeds (Podostemaceae) [22,82–84], but also in terrestrial plants such as the gloxinia family (Gesneriaceae), containing one-leaf plants (members of the genera *Monophyllaea* and *Streptocarpus*) [69,85] and in plant groups containing both aquatic and terrestrial members such as the bladderworts (*Utricularia,* Lentibulariaceae) [22]. The bladderworts (genus *Utricularia*) belong—also with respect to molecular developmental genetics—to the best-known examples of morphological misfits in vascular plants. The case studies presented in Sections 4.1–4.5 will show the need of continuum approaches and process thinking transcending classical plant morphology.

*3.3. Scientific Perspectivism and Complementarity in Plant Morphology*

> "If we once accept the fact that 'stem' and 'leaf' are no more than convenient descriptive terms, which should not be placed in antithesis as if they corresponded to sharply opposed morphological categories, the problem of their delimitation and of their differentiating characters vanishes into thin air." Agnes Arber 1930 [86] (pp. 308–309)

Baum [66] published a stimulating essay on the key question: **What *are* plant parts?** He focuses on the "dynamic nature of development" and claims—at least for multicellular plants such as vascular plants—that "parts are not sharply defined things like the bolts and nuts and gears of a car, but repeating eddies or vortices in developmental flows." Baum, as an enthusiastic proponent of EvoDevo, presented three complementary perspectives to understand the nature of plant parts such as leaves (or what botanists usually accept as "leaves" in vascular plants): (i) parts-as-structures (linking structures to their genetic causes), (ii) parts-as-functions (linking traits to responses to selection), and (iii) parts-as-processes (allowing to better recognize, e.g., "leaves" along a stem as being serially homologous). Baum [66] (p. 254) concluded: "Plant parts can properly be considered structures, functions, and processes".

**Four complementary views to look at leafy shoots in vascular plants.** According to Howard [87], the plant is a continuous whole. He emphasized this when he described the shoot of vascular plants as a "stem–node–leaf continuum." With regard to the leafy shoots in vascular plants, there are different perspectives that complement one another. During the two centuries since Goethe, the shoot of vascular plants was dismembered conceptually in (at least) four different ways [32,88,89]:

(I)　　The **classical stem-and-leaf model**: This is the popular model, as favored by most plant biologists, including classical plant morphologists (see Section 3.2). Stems (caulomes) and leaves (phyllomes) are accepted as structural categories that build up the shoot in seed plants and ferns. We may use this model as a "rule of thumb" (Tomlinson in Sattler [90]).

(II)　　The **fertile-leaf model**: This less known model accepts each leaf with one or more buds in its axils as one developmental unit, as part of one bifurcating meristem giving rise to both the subtending leaf and the axillary bud(s). This model, going back to Warming (1872) [91] and Goethe (1790) [92], explains why in seed plants (less so in ferns), lateral branches or flowers arise in the axil of leaves and may even form common primordia that bifurcate into a leaf and its axillary bud. Unequal bifurcation of such common primordia resulted in axillary flowers lacking their subtending leaves (nearly) completely, as it is typical for inflorescences in Arabidopsis and other members of the Brassicaceae [93]. Lyndon [94] (p. 21), as a proponent of this model wrote: "It is easy to forget that the leaf is not a single but a dual structure—a leaf with a bud in its axil." Shoot buds or flowers arising on leaves (a phenomenon known as epiphylly) provide further evidence for the validity (or at least the heuristic value) of the fertile-leaf model [95,96] (see Section 4.4).

(III)　　The **leaf-skin model**: In this model as proposed by Saunders [97] the terms leaf and stem are still accepted as structural categories, but the borderline between them is drawn differently. The stem cortex is conceived as being formed by the elongated leaf bases, which cover the stem core like a skin (see Section 4.3).

(IV)　　Unlike the above-mentioned three complementary models (I–III) that still accept leaf and stem as primary units of the shoot, each leaf or leaf whorl can be seen as a growth unit with the stem zone just below the node of its insertion, leading to the **phytomeric model**. In this shoot model, the stem zone, called **phytomere** (also written "phytomer"), consists of a leaf, its axillary bud, the node, and one internode below the leaf. There are many botanists and developmental biologists who favor this shoot model [26,27,32,93,98]. For example, Kinoshita and Tsukaya [69] wrote: "In the aboveground portion of a typical seed plant, a shoot is composed of the repetition of a unit called a phytomere consisting of a stem and a determinately growing leaf." Rohweder (1963) [99] and more recently Vita et al. [100] described and illustrated obvious phytomeres in growing shoots of herbaceous flowering plants such as Commelinaceae while focusing on leaf-related vascular tissue inside the stems, somewhat similar to a chain of inverted cones sticking into each other. The term phytomere is also applied to Arabidopsis, e.g., by Müller-Xing et al. [93]: "Phytomeres are metameric units that are composed of internode and node (leaf plus axillary meristem)."

Adherents of these four shoot models in vascular plants have often argued over which model is the correct or appropriate one [32]. Such arguments seem futile to a great extent. They are based on an either/or logic, meaning that any view is either correct or not. We can overcome this attitude by the acceptance of perspectivism and complementarity (see Section 2.2). Then, the different models complement one another, because they present different perspectives. Already Goethe [92] accepted different and contradictory perspectives on plant morphology. Among the four perspectives mentioned above, Goethe embraced three in different contexts, namely (I) the stem-and leaf model, (II) the fertile-leaf model, and (IV) the phytomeric model.

### 3.4. Historical Roots of Continuum Plant Morphology

As indicated above, the classical approach to plant morphology with a typological view of organ categories is no longer sufficient to explain all possible forms in flowering plants and ferns [6,19,49]. Rolf Sattler, as a young and enthusiastic botanist, regularly attended the conferences on plant morphology held in and around Germany. Fifty years ago (i.e., 1971) Sattler [101] gave an oral presentation on "A new shoot model", questioning the traditional view of German-style classical morphology (also called typology). Sattler was already professor of Botany and Biophilosophy at McGill University, Montreal (Canada). Because he had grown up in Germany and did all his studies (incl. PhD) mainly in Germany, he was well aware of classical plant morphology that was in the 1980s still a very strong botanical discipline in German-speaking countries, including Austria and Switzerland (see Section 3.2). Sattler [101,102] proposed that "leaves" and "stems" in vascular plants are no more clearly delimited structural categories; they can intergrade to considerable degree.

A historical survey leading from classical plant morphology (typology) to dynamic morphology (encompassing both continuum and process approaches) was published by Christian Lacroix et al. (2005) [49]. Sattler's school of continuum morphology indeed had its forerunners. For example, the principle of iteration (or identity-in-parallel) involving both shoot and leaf was formulated over 70 years ago by the British plant morphologist Agnes Arber. She was convinced that the principle of repetition is found everywhere in plants [28–30,49,86,103]. Focusing on vascular plants, Arber was puzzled by the fact that some steps of the developmental pathway of a whole shoot (i.e., leafy stem) can be repeated within a compound leaf. According to Arber [28] (p. 125) "a typical leaf is a shoot in which the apex is limited in its power of elongation and in its radiality." Thus, Arber [28,103] proposed the **partial-shoot theory of the leaf** in vascular plants [13]. Compound leaves may repeat in each part what they have already produced as a developing whole. Arber [28,103] described many examples of vascular plants showing repetition during growth and development. The repeated unit can be totally identical to the one already formed (i.e., complete repetition of a developmental pathway), or the structures formed afterwards may repeat the preceding ones only to some degree (i.e., partial repetition of a developmental pathway) while deviating in other features.

Honoring Agnes Arber, the continuum (and process) approach in plant morphology was labelled as **fuzzy Arberian morphology** (abbreviated "FAM") [13]. "Fuzzy" refers to fuzzy logic, "Arberian" to Agnes Arber. This approach is not only supported by many morphological data, but also by evidence from molecular genetics [72,104] (see Section 4.1). Arber herself also had forerunners. Several botanists in the 19th and the first half of the 20th century referred to intermediates between morphological categories and thus implicitly or explicitly recognized a continuum; see historical surveys in [13,88,102]. Unfortunately, they were often forgotten or ignored in mainstream plant morphology. Even Johann Wolfgang von Goethe [92], who designed the "Urpflanze" with its three main organs (roots, stems, leaves), accepted the basic tenet of Agnes Arber's partial-shoot theory of the leaf when he wrote: "When leaves divide, or rather when they advance from their original state to diversity, they are striving toward greater perfection, in the sense that each leaf has the intention of becoming a branch" (quoted by C. J. Engard in Mueller [105], p. 10).

Arber's [28,103] partial-shoot theory of the leaf parallels Minelli's [106,107] **paramorphism concept** for animal bauplans such as arthropods and vertebrates, according to which "animal appendages can be regarded as evolutionary duplicates of the main body axis, originated by a re-expression on a secondary (lateral) axis of the developmental routines responsible for the production of the main body axis... In terms of gross morphology, this is suggested for example by the segmented structure of the appendages in arthropods (and to some extent also in vertebrates), animals in which the main body axis is similarly segmented."

*3.5. Homology and Organ Identity in Classical and Continuum Plant Morphology: Acceptance of Partial Homology and Fuzzy Organ Identities*

**Homology versus convergence of plant structures:** Plant morphology is a comparative and evolutionary discipline, making comparisons between structures in plants that seem to be more or less related [108]. In groups of plants that are closely related, it is usually easy to explain similarities of plant structures as expression of shared ancestry and common genetics. Thus, similar structures are taken as being nearly identical ("homologous") to each other. In comparing distantly related groups of plants (e.g., ferns on one hand and flowering plants on the other hand), such a decision becomes more complex. Then similarity in appearance of certain plant structures may have evolved by convergence, usually an expression of independent adaptation to common environmental pressures.

The perspective of the researcher may determine whether certain bauplans and their modules are taken as homologous (synapomorphic) or seen as results of convergent evolution. For example, Harrison and Morrison [60], in an essay on the origin of vascular plant shoots and leaves, provided convincing evidence that the usually **planar foliage leaves evolved more than once** from three-dimensional branching shoot systems. This happened independently at least three times, in (1) lycophytes (clubmosses and allies) with "microphylls", (2) monilophytes (ferns, including horse-tails and whisk ferns) with "fronds", and (3) seed plants (gymnosperms and flowering plants) with "leaves" sensu stricto [109]. Why have leaves evolved multiple times? This question can be answered hypothetically by pointing to the environmental conditions during early phases of vascular plant evolution more than 400 million years ago [110,111]. Thus, Harrison and Morris [60] (p. 10) presented the following "survival of the fittest" hypothesis: "Polyphyletic leaf origins were coupled with declining atmospheric $CO_2$ levels, declining global temperatures, increasing stomatal and vein densities in leaves, the evolution of extensive rooting systems, and increasing plant competition for space to acquire environmental resources."

**What is homology in a comparative biological context?** Conceptual realism (essentialism, substance ontology) in biology often resurface in more or less subtle forms as either/or questions. Such questions are often framed in terms of homology. Homology has been called "morphology's central conception" [112]. Originally, it was defined as "essential similarity, which implied 1:1 correspondence" [80,113–115]. With regard to animals, Owen (1843) [116] (p. 379) defined a homologue as "the same organ in different animals under every variety of form and function". So, how can we determine what is the same organ? Traditionally, three criteria were used to decide if two structures in animals or plants are homologous to each other or not [56,108,117–120]. Owen used the criterion of **relative position** with respect to other structures = topological similarity, including correspondence in developmental origin (1st criterion). Homology may also be defined by the 2nd criterion of **special quality** (i.e., similarity in structural details, including anatomy, physiology, organ functions). Besides the first two criteria (position, special quality), the 3rd criterion of **transitional forms** is often used, emphasizing a transformation series of intermediates between putatively homologous structures. With the advent of developmental genetics, the "molecular players", i.e., **gene expression** behind complex structures or organs (phenotype), may be used as a 4th homology criterion: Homologues often do share developmental pathways, which are based on common gene pathways, especially when related groups of organisms are compared [49,72,104,120–125].

There is no doubt that the study of developmental genetics has added a great deal to our understanding of morphological evolution. If we try to accept 1:1 correspondence between morphological structures and gene functions, it may be useful to follow the four-steps recipe offered by Jaramillo and Kramer [124]: "In order to make comparative expression analyses more meaningful, we need to take into account (1) the evolutionary history of the morphological feature in question through reconstructions on robust phylogenies; (2) the evolutionary history of the candidate gene, because gene duplications are very common ( . . . ) and these events may make the occurrence of neo- and sub-functionalization more common among plants than other organisms; (3) a clear definition of the morphological character of interest; in the case of flowers—perhaps the best-understood angiosperm

structures—it is important to distinguish each whorl of organs as a distinct character and the identity that they can take (sepaloid organ, petaloid organ, stamen, carpel) as a character state; (4) comparative expression data from multiple genes in a genetic network responsible for the character of interest [121]." (For details, see case study on flowers in Section 4.4).

However, the homology concept as used in animals and plants is a very difficult issue. There are five different levels of homology in biological systems, from serial (repetitive) homology within a modular organism, to historical homology (also called synapomorphy) down to underlying (=latent) homology and deep evolutionary homology at the molecular genetic level [62,108,126,127]. Therefore, Brigandt [120,123] accepted homology as something like a "nomadic concept" (see Minelli [4]): "Once it migrated from comparative anatomy into new biological fields, the homology concept changed in accordance with the theoretical aims and interests of these disciplines."

**Organ identity and meristem identity:** An organ in multicellular animals and plants is a part of a living organism with a certain set of functions besides its positional and constructional characters. In the context of plant developmental genetics, "organ identity" means the developmental fate of an uncommitted primordium. This term is used in zoology (e.g., [128]) and botany as well. Organ identity can be defined by morphological criteria and by their gene expression pattern, including organ identity genes that sculpt, for instance, the structure of angiospermous flowers (see Section 4.4). The organ identity concept is closely related to the concept of homology; both have multiple and sometimes conflicting meanings, as reviewed by [35,108,119,121,122]. Organ identity as a concept is also used outside the floral region. The vegetative body of vascular plants shows primordia that are committed to a certain developmental fate [109,129] (see Section 4.2). Acquisition of organ identity often happens progressively rather than at once [130]. In ferns and aberrant flowering plants (morphological misfits) such as *Utricularia,* the commitment of a primordium to become a leaf or shoot (including stem) can be considerably delayed [13,131] (see Sections 4.1 and 4.3). The term "identity" is also used for meristems such as shoot apical meristems (SAMs) in vascular plants that change their identity during the development of an individual plant from seedling stage to flowering. Müller-Xing et al. [93] wrote with respect to Arabidopsis: "The identity of the SAM undergoes several changes during the plant's lifecycle and so do the generated organs that cause modifications in the shoot structure, which can be described by metameric units named phytomeres." Thus, the term "meristem identity" characterizes the growth phases of a shoot apical meristem (SAM), with vegetative meristem, inflorescence meristem, and floral meristem as three possible "identities" [35,93,132].

**Partial (=mixed) homology as useful concept for both plants and animals:** Homology also includes partial homology, as proposed for vascular plants by Sattler [19,113,119,133]. The continuum approach in plant morphology is based on the acceptance of partial homology, as reviewed by Sattler [7]: "I proposed a partial homology concept (as a semi-quantitative homology concept) over 50 years ago (in 1966), but it was not well received by many typologists and evolutionary morphologists who, like typologists, insisted that all homology must be total, that is, 1:1 correspondence (or essential similarity). Partial homology leads in vascular plants to a continuum between structural categories (plant organs) such as, leaves, stems, roots, and even multicellular hairs/trichomes." It seems that the concept of partial or mixed homology is more frequently applied to plants than to animals [56,107]. However, Minelli [80] wrote also with respect to animals: "Taking distance from the traditional, all-or-nothing approach according to which two structures are either homologous or nonhomologous, since the 80s of the past century, several authors have been defending the view that all assessments of homology are by necessity partial." Research in EvoDevo has provided further corroboration for this view [114]. Now we have much evidence for partial homologies, even at the molecular level (see Section 3.8).

*3.6. Process Plant Morphology and Morphospace in an Evolutionary Context*

"I believe the apparent failure of botanists to internalize process morphology arises because this philosophy requires habits of thought and ways of communicating that are not natural

for science in general (Seibt [33]). Due to these possibly hard-wired psychological biases, it is very hard to resist the tendency to perceive plants as objects, drawing us to a language of nouns not verbs." David Baum (2019) [66].

According to classical plant morphology ("typology") sensu Wilhelm Troll [65] and Donald Kaplan [67,68], a plant structure is either this or that, but not both. This classical approach is widely used in the study of model plants with typical organs [13,49] (see Section 3.1). Continuum plant morphology allows the acceptance of intermediates and mixed identities between structural categories (see Section 3.5 above). Both classical plant morphology and continuum plant morphology have shortcomings, because both depend on the recognition and definition (crisp or fuzzy) of structural categories. Process morphology (also called dynamic morphology [49]) may be a way of overcoming these shortcomings, because it allows the replacement of structural categories by process combinations ("developmental routines").

Leaf, stem, and root are often—unconsciously—taken for granted as organs in vascular plants. When we realize that these structural categories are, to some extent, arbitrary concepts, each of them encompassing a certain set of developmental processes, then we are prepared to abandon structural concepts and instead refer to combinations of developmental (morphogenetic) processes that depend—to some degree—on gene regulatory networks. This radical view was proposed by members of the morphological school of Rolf Sattler (including Barabé, Jeune, Lacroix, and Rutishauser), as well as Baum and Langdale [1,6,13,19,49,66,119,134–139].

**Theoretical and empirical morphospace.** The living forms we perceive and conceive of in the realms of multicellular organisms (animals, plants, fungi) "are only a small subset of the possible forms we could imagine" (Minelli [37]). The theoretical morphospace includes all possible process combinations for seed plants, whereas the empirical morphospace contains only those process combinations that are realized in nature [49,140]. Each axis of the morphospace corresponds to a variable that describes some developmental processes of an organism, or its parts. The use of a single morphospace to which gene expression can be annotated is appealing, especially because its use would remove some terminological problems described above.

Process morphology allows seeing whole plants as combinations of developmental processes instead of more or less arbitrarily assigning plant parts to categories such as root, stem, and leaf. Parameters for process morphology with respect to vascular plant development are growth duration, symmetrization (e.g., acquisition of dorsoventral symmetry), positioning, branching, tropism, rhythm, reiteration, and senescence (see [49,141] for lists of the main processes in plant development).

We may ask with respect to morphological misfits such as bladderworts (*Utricularia*) and river-weeds (Podostemaceae) having unusual morphologies [22]: Is the recognition of developmental processes (e.g., branching patterns and growth patterns) more important than proper definition of structural units, i.e., plant organs such as roots, stems, and leaves? According to Sattler [7] "there are no misfits according to process morphology; everything fits because everything can be understood in terms of branching and articulation. By using developmental processes instead of structural categories, morphological misfits such as bladderworts and river-weeds need no longer be forced into one or the other category."

**Hypotheses on bauplan evolution in vascular plants:** Schneider [74] addressed the evolution of early land plants (ferns mainly), while focusing on the transformation of structures in time [15,142]. This strictly phylogenetic approach is more likely to be compatible with the developmental, process-oriented perspective of fuzzy Arberian morphology than with the typological classification of classical plant morphology [13,15]. The importance of "transformation of forms" versus "fixed typology" can be illustrated with the discussion on the evolution of leaves. The origin of leaves is frequently discussed, and the nonhomology of "leaves" of liverworts, mosses, lycophytes, ferns, and seed plants is widely accepted (see Section 3.5). However, these statements do not provide us with any understanding of the origin of leaves. Process combinations ("developmental routines") are canalized during land plant evolution [15,53,143–145]. Therefore, a process morphological

approach may help understanding the evolutionary origin of bauplans in vascular plants. Walter Zimmermann's [53,143] "telome theory" seems to be still useful as an evolutionary hypothesis in the sense of process morphology, because it proposes a sequence of transformative evolutionary processes, especially (i) unequal branching (overtopping), (ii) rearrangement of lateral branches into a single plane (planation), and (iii) infilling of spaces between branches with laminar tissue (webbing/fusion). According to the telome theory, a dichotomously branching shoot system evolved into a foliage leaf by these three transformations [60,142]. Although Zimmermann's telome theory as evolutionary hypothesis has serious limitations, it is still discussed as a heuristic aid by developmental geneticists in search for the regulation of mechanisms of developmental processes (e.g., [144–147]. It is a gradualistic perspective that breaks the evolutionary origin of complex organs such as fern leaves (fronds) into steps (see Section 4.1, also Jill Harrison's Blog 2018: https://thenode.biologists.com/testing-zimmermanns-telome-theory/ research/).

The step-by-step evolution of leaves (fronds) as proposed by Zimmermann coincides with what was called partial or mixed homology (see Section 3.5) because leaves/fronds probably evolved by a sequence of evolutionary transformations. Of course, these hypothetical transformations in the sense of Zimmermann's telome theory need to be understood in terms of the genetic regulation of plant developmental processes [60,144,148]. According to Schneider [74]: "A process-oriented approach, combining phylogenetic and developmental perspectives, instead of a typological-oriented approach, is the most promising approach to identify the developmental pathways and their transformation in the evolution of land plants." Zimmermann [143] (p. 470) was already an early proponent of process morphology when he wrote: "The telome theory not only applies to 'typical' organs that means to such organs as are classified according to classical morphology under the notion of leaf, shoot, axis, stem, root—but all forms of 'Kormophyta' (i.e., vascular plants, including ferns and seed-plants) are to be traced back to the varying combinations and the different degree of manifestation of a few modifying processes."

Process thinking may even allow to define plant structures such as "shoots" differently (if such a definition is needed at all). Plackett et al. [109] (p. 4) argued about the evolution of land plant shoots as follows: "From a developmental perspective, canonical plant shoots (as generally recognized in vascular plants) can be defined as a process, i.e., they develop iteratively from an apex to produce lateral organs. Using this definition, the gametophores of extant mosses and 'leafy' liverworts can also be classified as shoots, possessing an axial body-plan." This growth behavior contrasts with the "thalloid" bauplan of other liverworts (e.g., *Marchantia*), which possess a usually flattened plant body (comprising several cell layers) with apical growth.

Gilbert and Bolker [149] distinguished between process homology (i.e., homology between developmental processes) and structural homology, which represent different hierarchical levels of homology [107]. Jaramillo and Kramer [124] (p. 69) wrote: "Process homology reflects the common inheritance of developmental genetic pathways or modules that can be co-opted to function in diverse situations. For this reason, process homology is dissociable from structural homology. ... Novel combinations of such genetic modules may be at the heart of many of Sattler's ([113,119]) examples of partial homology or may promote the evolution of novel organ identities."

Several developmental geneticists are aware of a certain degree of fuzziness in plant development. They have used fuzzy concepts such as "leaf-shoot indistinction" [150], "leaf-shoot continuum model" [151,152], and "mixed shoot-leaf identity" [153]) to describe unusual plant structures. These fuzzy concepts are consistent (or nearly so) with fuzzy Arberian morphology, including both continuum and process morphology [72,104]. James [154] summarized as follows: "It is now widely accepted that ... radiality (characteristic of most shoots) and dorsiventrality (characteristic of leaves) are but extremes of a continuous spectrum. In fact, it is simply the timing of the *KNOX* gene expression!" With respect to developmental similarities between leaves and shoots in vascular plants (including Arabidopsis), Prusinkiewicz and Runions [155] gave credit to Agnes Arber: "This similarity is consistent with the 'partial shoot theory' (Arber [28]), which emphasizes parallels between

the growth of shoots and leaves. The strikingly different appearance of spiral phyllotactic patterns and leaves would thus result not from fundamentally different morphogenetic processes, but from different geometries on which they operate: an approximately paraboloid SAM dynamically maintaining its form vs. a flattening leaf that changes its shape and size."

### 3.7. Mathematical Tools as a Test for Continuum and Process Plant Morphology

In plant morphology, most structures of vascular plants can easily be assigned to pre-established organ categories. However, there are morphological misfits, i.e., intermediate or deviating structures that do not fit those categories associated with a classical approach to morphology (see Section 3.2.). With the mathematical tool of principal component analysis (PCA), Jeune, Lacroix et al. [138,139,156,157] found a way to unify classical and continuum plant morphology as a synthesis through the space of forms. To integrate the diversity of forms in the same general framework, they constructed a theoretical morphospace based on a variety of modalities, where it is possible to calculate the morphological distance between plant organs. Concentrating on shoot, leaf, leaflet, and trichomes as structural categories (but ignoring the root) Jeune, Lacroix et al. tested the hypothesis that classical morphology (typology) and continuum morphology occupy the same theoretical morphospace. They concluded that the relationship between the two approaches remains a question of weighting of criteria. Typical morphological elements (e.g., shoots, leaves, trichomes), atypical structures (e.g., leaf-like shoots, called "phylloclades"), and particular cases representing exotic structures such as the epiphyllous appendages of *Begonia* and "water shoot" and "leaf" of aquatic bladderworts (*Utricularia*) were placed in the morphospace. The more an organ differs from a typical shoot, the further away it will be from the barycentre of shoots. By giving a higher weight to variables used in classical typology, the different organ categories appear to be separate, as expected. If we do not make any particular arbitrary choice in terms of character weighting, as it is the case in the context of continuum morphology, the clear separation between organs is replaced by a continuum. By using an equal weighting of characters, contradictions due to the ponderation of characters are avoided. Thus, Jeune, Lacroix et al. [138,139,156,157] tend to merge classical plant morphology (mainstream morphology, based on a qualitative homology concept implying mutually exclusive categories) and continuum morphology into the more encompassing process morphology = dynamic morphology (see Section 3.6).

### 3.8. Possible Lack of One-to-One Correspondence Between Structural Categories and Gene Expression in Higher Plants

"We are still far away from understanding how three-dimensional forms are generated by the genetic system." Diethard Tautz (2019) [158]

We live in a gene-centric world. Many biologists accept the view that the development of living organisms (animals, plants, fungi, ... ) is controlled mainly by genes. Genetic determinism also governs the thinking of many laymen and journalists with respect to heritability of human health and diseases.

However, we have to consider that the genome (DNA) in the cells of living organisms is a vital but not the only driver of developmental processes, as expressed by Gilbert and Bard [159]: "The fertilized egg inherits DNA; it does not inherit 'genes.' Genes and gene products are constructed anew in each cell in the developing embryo by the relationships between DNA, transcription factors, and RNA-splicing factors. Only certain regions of the genome can be genes in different cell types." For similar statements see also [56,158].

We must know what an organ is before we can investigate gene expression in that organ. If structural categories do not provide adequate descriptions of plant structure, perhaps it is possible to define structures based on developmental genetics. This new way to find homology between structural units was already presented as the 4th homology criterion (see Section 3.5). If there is a one-to-one correspondence between structural units (e.g., roots, leaves, and flowers) and the "molecular players behind the characters" [125], it should be possible to identify the structural units by the expression

of well-characterized marker genes. To do this, we need to look for, e.g., **organ identity genes** in order to define the structural categories clearly. For example, the *KNOX/ARP* module (as used by Katayama et al. [82,83], Cruz et al. [72,104]) helps with the identification of the leaf as a determinate unit, and the shoot as an indeterminate module in vascular plants. This approach seems to have promise in the cases where control genes for organ identity have been shown to exist, for example, Pax6, which is often accepted as the "master control gene" for eye development in arthropods and vertebrates [57,160].

There are, however, various difficulties when we try to define plant structures by a set of expressed genes. Adopting this approach for all plant organs leads to some bizarre conclusions, because structural homologues in biological systems do not always have the same underlying molecular genetic machinery [62]. Jaramillo and Kramer [124] present arguments against a too strong belief in genetic determinism, especially in combination with homology: "Genetic information is not always a reliable indicator of homology, especially when only one gene is examined. Our particular concern is the conflation of positional evidence for homology with evidence from identity. Many of the genes that have been studied comparatively are floral organ identity homologs, and it appears that such identity programs can be shifted spatially, resulting in homeosis." Examples from both botany and zoology show that homologous structures can result from different genetic controls [62,107,127,161,162]. For instance, leaflet formation in pinnately compound leaves of many eudicots is correlated with *KNOX1* expression in leaf primordia [163,164]. In contrast, in legumes (Fabaceae) such as pea (*Pisum sativum*), the formation of compound leaves depends on the expression of the PEAFLO gene, the pea homologue of LEAFY from Arabidopsis [152,165].

The lack of correspondence may be due to imprecise morphological homology assessments, but it may also arise from the reuse of existing genetic resources in novel contexts because transcription and signaling factors are often used multiple times in context-specific combinations within an organism [34,166]. Vergara-Silva [137] (p. 260) concluded similarly: "Distinct groups of genes that in principle act in one categorical structure are also expressed in another, and the consequence that this overlapping pattern has on cell differentiation is an effective blurring of the phenotypic boundary between the structures themselves." Developmental genetics underlying morphological continua in flowers, and interesting cases of flower–inflorescence continua will be discussed in Section 4.4 (see [132]).

### 3.9. The Explanatory Power of Process Morphology in Multicellular Plants

Various plant biologists seem to be aware of the explanatory power of process morphology. Much architectural diversity in vascular plants results from the varied growth patterns of apical and axillary meristems [167]. Computational biologists and mathematicians such as Prusinkiewicz and colleagues [155,168] provided further evidence for the heuristic value of continuum and process plant morphology. As mentioned by Prusinkiewicz and Runions [155]: "The form of plants is not coded directly in their DNA, but is produced by a hierarchy of developmental processes that link molecular-level phenomena to macroscopic forms. Many of these processes have a self-organizing character, which means that forms and patterns emerge from interactions between components of the whole system." For example, a suite of developmental processes leads from the leaf primordium to the mature leaf. Tsukaya and his collaborators [85,169,170] subdivided leaf morphogenesis of Arabidopsis, Antirrhinum, and other angiosperms into subsequent processes (phases). They admitted that the mechanisms regulating each process of morphogenesis, such as leaf determination, establishment of dorsoventrality (e.g., adaxialisation), and polarity recognition, are still badly known. Molecular genetics allows to approach to such processes and relevant developmental pathways that are influenced but not completely controlled by gene networks. Molecular genetic studies in recent decades have considerably expanded our understanding of the mechanism of leaf development (including adaxial–abaxial patterning and lamina flattening) in angiosperms [85,169,170]. Combinations of developmental processes even allow to transcend the leaf–stem (or leaf–shoot) boundary. For example,

Nakayama et al. [171] analyzed in *Asparagus* spp. the developmental genetics of the photosynthetic needle-like organs (labelled as "cladodes") that are—according to classical plant morphology—modified shoots (stems). They observed in developing *Asparagus* cladodes the expression of both SAM-related and leaf-lamina-related genes [85]. This result coincides with comparative developmental studies by Cooney-Sovetts and Sattler [172], who interpreted the leaf-like shoots ("cladodes", "phylloclades") in *Asparagus* and allied genera (*Ruscus, Semele*) in the monocot family Asparagaceae as homeotic mosaics, combining developmental processes of both leaves and shoots (stems).

## 4. Case Studies: Developmental Genetic Studies Supporting the Continuum View in Plant Architecture

The case studies mainly focus on morphological misfits that do not obey the bauplan rules of classical plant morphology (as introduced above, especially Section 3.2). The case studies point to plant structures that are difficult to explain by a simple one-to-one correspondence between structure and gene function. The genetic study of these organisms will likely reveal that some of the phenotypic fuzziness results from overlapping developmental programs (compare Section 3.5, Section 3.6, Section 3.7, and Section 3.8).

### 4.1. Ferns: Continuum Between Compound Leaves (Fronds) and Shoots

Many botanists accept the hypothesis that during land plant evolution, megaphylls (including fronds and prefronds) of ferns were derived from branched axial organs ("telome trusses") by flattening ("planation"), webbing ("fusion"), and a switch to determinate growth (see Section 3.6). Prefronds, as found in early vascular plants such as the extinct progymnosperms (e.g., *Archaeopteris*), are three-dimensionally branched structures and are seen as phylogenetic precursors of compound leaves, especially fronds of modern ferns [15,53,143,167,173].

Comparative morphologists also studied the developmental patterns of modern-day ferns. Leaves of ferns are often labelled as "fronds", because they have an evolutionary history different from the foliage leaves of seed plants [109,174] (see Section 3.5). Developmental mosaics between leaves and shoots are described for various recent ferns, e.g., *Gonocormus* (*Trichomanes* group) with shoot buds developing from the rachis [175]. To label a group of embryonic cells derived from the shoot apical meristem (SAM) as "leaf" is to label it according to its presumptive developmental fate (organ identity) under undisturbed conditions. However, surgical or chemical treatment may change the developmental fate of what was supposed to become a leaf. In some ferns at least (e.g., species of *Dryopteris, Hypolepis, Osmunda*), such experimentally induced changes include the developmental switch from a leaf (frond) primordium to a shoot (rhizome) bud and vice versa [176,177].

**"Transformation of forms"** versus "fixed typology" in the evolution of fern leaves (fronds). Schneider [74] presented a remarkable essay on the evolutionary morphology of ferns (*monilophytes*), while employing a process-oriented approach, which combines the process thinking of the fuzzy Arberian morphology with the process orientation of Darwinian evolution (see Section 3.6). Schneider [74] stressed the heuristic value of continuum plant morphology that employs a holistic view of the plant body in which the different organs, such as leaves, shoots, and roots are linked by shared developmental processes. In comparison to angiosperms, the leaves (fronds) of ferns may share a higher degree of similarity with shoots. Thus, the differentiation between the organs may be understood as less distinct. For example, the leaves of ferns often show a long-lasting apical growth that may be caused by a similar expression of transcription factors in the shoot apical meristems (SAMs) and the leaf apical meristems (LAMs). In addition, several unusual fern structures, for example, stolons of *Nephrolepis*, can be best described as misfits that combine features of different organs [72,74,104,148,174,178,179].

New EvoDevo studies on ferns point to the remarkable similarity of compound leaves (fronds) and shoots [72,104,177]. Raphael Cruz et al. [72] (p. 2) wrote: "Fern leaves are different from most seed plant leaves. For example, unlike seed plants, many fern leaves have a leaf apical meristem (LAM). In ferns, the LAM is responsible for a transient indeterminacy during leaf development, usually producing

pinnae during a longer period than the regular compound leaf of a seed plant . . . Fern leaves resemble the indeterminate shoot by having an apical meristem, producing lateral organs and having a transient or even persistent indeterminacy (as in the genera *Lygodium, Nephrolepis, Salpichlaena, Jamesonia*), as reviewed in Vasco et al. [180]. These features do not fit the classical morphological concept of leaves as they do for seed plants." Based on developmental genetic studies on compound leaves in vascular plants [181], there is evidence that *Class I KNOX* genes are directly associated with indeterminacy and are required to make compound leaves in many cases, representing a partial homology with the shoot. *Class I KNOX* genes are also an important marker of meristematic activity in fern shoots [71]. Cruz et al. [72] found *Class I KNOX* expression in SAMs as well as LAMs (and even in growing pinna tips) of *Mickelia scandens.* Their conclusion: "Fern apical meristems should be interpreted as a complex and highly organized interconnected network of cells with indeterminate fates, specialized zones (apical cells vs. peripheral cells), and the capacity for producing new organs (leaves or pinnae)." According to Cruz et al. [72,104], Harrison and Morris [60], Harrison et al. [148], Plackett et al. [109], and Vasco and Ambrose [177] fern leaves and their segments have to be interpreted evolutionarily and ontogenetically as reduced shoots. Thus, the presence of shoot-like features in developing leaves is strong evidence in favor of Agnes Arber's partial shoot theory (see Section 3.4). Cruz et al. [72] (p. 9) stressed the importance of Arber's ideas with respect to future studies in developmental genetics: "Her theory should be strongly discussed now that new molecular evidence, as our results and other studies discussed here, is available. Our results point to multicellular meristematic structures in the shoot, leaf, and pinna apices, also reinforcing her idea of "identity-in-parallel", in which structures may be put in a relation of the part to the whole, but is also equivalent as a whole. The pinna is part of the shoot, but ultimately is equivalent to a whole shoot, carrying the potential of producing new lateral structures."

*4.2. Flowering Plants—Leaf and Shoot Development in Flowering Plants*

**Compound leaves with determinate apical growth**. Foliage leaves in most flowering plants are determinate organs: together with the presence of an axillary bud, this is indeed one of the criteria commonly used to distinguish a "true" leaf from a leaf-like branch (see Section 3.1). Leaves of most flowering plants are initiated sequentially as transversely inserted primordia. They arise at a shoot apical meristem (SAM), which usually shows an indeterminate growth [13,85,176,182]. Similarly, young compound leaves of angiosperm genera such as celery (*Apium,* (Apiaceae), sumac (*Rhus*, Anacardiaceae), neem (*Azadirachta*, Meliaceae), American river-weed (*Marathrum*, Podostemaceae), and orange jasmin (*Murraya*, Rutaceae) are provided with a meristematic leaf tip = leaf apical meristem (LAM). These taxa show an acropetal mode of initiation with leaflet primordia, which are inserted transversely at the LAM. Thus, they resemble leaf initiation at the SAM [49,139,183–187]. — Bharathan and Sinha [188] (p. 1533) summarized while focusing on flowering plants: "The true homologies of compound leaves have been a matter of debate. They have been considered true lateral organs with homologies to simple leaves [189,190], or structures that are intermediate between leaves and shoots [184,191]." Despite the general open-mindedness towards shoot-like features in compound leaves [85,192], there are still various developmental geneticists [193,194] who search for plant hormones and transcriptional regulators modulating compound leaf development—without having the "identity-in-parallel" between leaves and shoots in mind that was formulated by Agnes Arber [28] (p. 142) more than seventy years ago: "To the compound leaf, the leaflet stands in relation of *part* to *whole*, but it is also the equivalent of the compound leaf as a *whole*, though in another generation."

**Compound leaves with indeterminate apical growth.** Flowering plants have two types of indeterminate leaves: those with a long-lasting apical meristem and those with a long-lasting basal meristem (see below). Indeterminate leaves with a long-lasting leaf apical meristem (LAM), as observable in many ferns (see Section 4.1), exist only in a few groups of flowering plants. Their meristematic activity is maintained for months or even years [56,85,169]. Field studies in woody members of the mahogany family (Meliaceae), especially *Chisocheton* and *Guarea,* have revealed that

their compound leaves may show seasonal apical growth for several years. The LAM contributes new leaflets in a manner similar to the SAM [176,191,192,195–197]. Both shoot tips and leaf tips of *Chisocheton* and *Guarea* species show seasonal flushing. Each time the SAM adds new leaves, the LAM also initiates new leaflets. Thus, the meristematic leaf tips and the meristematic shoot tips react to the same endogenous or environmental stimuli. Moreover, the leaf axis (i.e., petiole and rachis) increases in diameter due to secondary thickening with wood production, similar to what is typical for stems (shoot axes) of woody plants. In addition, certain species of *Chisocheton* produce epiphyllous shoots (*Ch. tennis*) or epiphyllous inflorescences (*Ch. pohlianus*) [198]. Thus, compound leaves of *Chisocheton* and *Guarea* have developmental routines resembling whole shoots.

**Stipules repeat the developmental pathways of whole leaves.** Stipules are basal outgrowths of foliage leaves. Quite often the presence and the typical arrangement of stipules is characteristic for a whole group of plants. Flowering plants such as the rose, pea, and coffee families (Rosaceae, Fabaceae, Rubiaceae) are characterized by stipules [13,192,199–201]. The Plant Ontology Consortium (www.Plantontology.org) provides the following definition for stipule (PO:0020041): "A cardinal organ part that is an appendage at the base of a vascular leaf. Usually one of a pair of appendages. May take the form of a spine (PO:0025174) or may be bristled or brush-like. Stipules may occur on the opposite side of the shoot axis from the leaf they are associated with, but they are still thought to arise from part of the leaf." Leaf blade and stipules usually arise from a common primordial bulge at the shoot apex, but stipules often behave quite independently from the leaf to which they belong. They usually stop growth clearly before the leaf blade itself is mature. They often abscise much earlier than the leaf blade. However, in various flowering plants such as woodruff and allies (*Galium* and related genera in Rubiaceae) leaf and stipule look the same or nearly so, indicating that the stipule is replaced by a leaf. In the flamboyant tree (*Delonix regia*, Fabaceae) the stipules repeat the branching pattern of the much larger compound leaf blade. Various examples of ectopic expression of leaf identity in stipular position are described in [35,192,199]. Arber [28] (pp. 80–82) concluded: "The relation of the stipules to the leaves of which they are members, may be compared with the relation of cotyledons to the primary shoot, and of prophylls or bracteoles to a lateral shoot."

Developmental genetics may help in defining "leaf identity" versus "stipule identity". A homeotic replacement of the stipule by a leaf was described in pea (*Pisum sativum*) mutants such as *cochleata* [202]. There are tendrils forming the blades as well as tendrils arising from stipular positions, a situation not known from any wild-type member of Fabaceae [129,152,203]. In the *afila* (af) mutant, all primary pinnae are replaced by a bunch of tendrils, whereas the stipules are not altered. A gene known to interact with the *af* gene is *sinuate leaf* (*sil*), which results in undulating margins of both leaflets and stipules. When combined with *af*, *sil* plants have adventitious tendrils arising from clefts in the distal portion of the stipule [35,203]. Thus, pertinent characters of the leaf blade can be ectopically expressed in stipular sites. Yaxley et al. [202] wrote that these mutants "change stipules into a more 'compound leaf-like' identity".

Stipules in flowering plants are, by definition, restricted to the leaf base. However, a few mutants in Arabidopsis and pea are known to have supernumerary stipules, which are ectopically expressed as part of the leaf blade or rachis [204]. The so-called "stipels" at the base of the lateral leaflets in compound leaves of the garden bean (*Phaseolus vulgaris*) may be understood as ectopically expressed supernumerary stipules, repeating the developmental leaf program at the base of its subunits [13,28].

**Indeterminate meristematic activity at the leaf base in one-leaf plants (*Monophyllaea*, *Streptocarpus*).** Besides leaves that grow indeterminately at their tip (as typical for ferns and few flowering plants) there are one-leaf plants having only one large foliage leaf at maturity. Because of this bauplan oddity, they were called "morphological misfits" [75] (see Section 3.2). The only foliage leaf per plant equals one of the two cotyledons that grows indeterminately due to the long-lasting meristematic activity at the base of the lamina [56,85,205,206]. One-leaf plants mainly belong to the genera *Monophyllaea* and *Streptocarpus* in the Gesneriaceae, which is a family of asterids in eudicots [207]. Unlike typical vascular plants, one-leaf plants do not have a typical shoot apical meristem (SAM).

The developmental biology of the one-leaf plants (*Monophyllaea, Streptocarpus* spp.) was recently studied by Kinoshita and Tsukaya [69,208]. The indeterminate growth of the only foliage leaf in these plants is supported by a groove meristem. Gene expression and physiological analyses proved that the groove meristem is equivalent to the SAM of typical flowering plants. Using classical terms for structural units in the one-leaf plants may be problematic. In *Streptocarpus,* there are several species with a usual bauplan, with elongate stems carrying several paired leaves. Somewhat comparable with the phytomeric shoot model (see Section 3.3), Jong and Burtt [205] proposed the phyllomorph concept fifty years ago in order to describe the unusual structure of the one-leaf plants (e.g., *Streptocarpus wendlandii*) lacking an obvious stem. The "phyllomorph" is defined as a novel but fuzzy developmental module consisting of an indeterminately growing leaf lamina and a "stem- and petiole-like structure" (called petiolode) at its base. Thus, the phyllomorph of a one-leaf plant corresponds to a shoot (including stem) in typical plants [69,85,208]. For a recent review on *Streptocarpus* phyllomorphs including regulation of meristem activity by plant hormones, see Nishii et al. [209].

**"Stem-leaf mixed organs" in river-weeds (Podostemaceae).** River-weeds are peculiar flowering plants that are adapted to river-rapids and waterfalls in the tropics. They are related to St John's wort (*Hypericum*) within the eudicots. Bell [75] called the river-weeds "morphological misfits", because they follow structural rules that are different from those of more typical flowering plants (see Section 3.2). The boundaries between organs known as roots, stems, and leaves—distinguishable in related flowering plants – are blurred (fuzzy). However, the members of this family have stable floral bauplans [22]. When we cling to structural categories such as leaf, stem, and root for the description of the plant bodies in river-weeds, we get into trouble with either/or homology ("sameness") of the various plant parts. Then we are forced to accept the existence of structural intermediates such as "stem-leaf mixed organs" in Podostemaceae, as genetically analyzed by Katayama et al. [82,83]. This appellation is meant to indicate that these structures have some features of leaves, and some of stems, probably due to their unusual gene expression pattern and the lack of obvious SAMs [84]. Summarizing the developmental genetic results in Podostemaceae and other flowering plants, Eckardt and Baum [210] concluded: "It is now generally accepted that compound leaves express both leaf and shoot properties."

*4.3. Flowering Plants—Bladderworts and Allies (Lentibulariaceae)*

> "It is probably best, as a purely provisional hypothesis, to accept the view that the vegetative body of Utricularias partakes of both stem nature and leaf nature. How such a condition can have arisen, historically, from an ancestor possessing well-defined stem and leaf organs, remains one of the unsolved mysteries of phylogeny." Agnes Arber (1920, p. 107) [211]

The case studies above (Sections 4.1 and 4.2) have shown that leaves of vascular plants can have indeterminate growth. They can even have terminal buds. This case study covers the bladderworts (*Utricularia* and allies), plants with highly unusual architectural rules deviating considerably from the body plan of typical flowering plants. Minelli [56] (p. 254) summarized the puzzling situation found in bladderworts properly: "Mechanisms specifying the identity of individual organs are unusually plastic: of the various primordia in a rosette, some will become leaves, others stolons, but their fate can be changed even at a very late stage of development. In some species, a stolon can eventually stop growing, completing its activity with the production of a terminal leaf. On the other hand, the tip of a nearly mature leaf can continue growing and turn into a new stolon." It is difficult to describe and interpret the developmental architecture of the bladderworts in terms of classical plant morphology. Continuum approach and process thinking provide complementary perspectives if we really want to understand the morphological oddities in this group (see Sections 3.5–3.7). In their flowers, however, the bladderworts behave as typical flowering plants, showing that they belong to the asterid eudicots [22,75,212].

The Lentibulariaceae are a family of flowering plants showing carnivory. They attracted and still attracts plant scientists, mainly because of the unique features in biology, morphology, and developmental genetics. The three genera in this family can be distinguished by their specialized traps for carnivory:

flypaper traps (passive) in *Pinguicula*, eel traps = lobsterpot traps (mainly passive) in *Genlisea*, and bladder traps (that conduct suction in < 1ms) in *Utricularia*. The genus *Utricularia* (bladderworts) is with >230 species by far the largest of the three genera. These plants live fully immersed in water (*U. australis, U. gibba, U. vulgaris*), on wet soil (e.g., *U. dichotoma, U. livida, U. reniformis, U. sandersonii, U. longifolia*), or even epiphytic (e.g., *U. alpina,* again *U. reniformis*) [212–214]. Irrespective of their ecological preferences, all *Utricularia* species are provided with tiny bladders used as active sucking traps for small animal prey in water.

The Lentibulariaceae have attained recent attention because of their dynamic nuclear genome size. Their genomes span an order of magnitude and include species with the smallest genomes in angiosperms, making them a powerful system to study the mechanisms of genome expansion and contraction. Moreover, developmental geneticists started to reveal the genetic network behind the morphogenesis of the peculiar traps in *Utricularia* [215,216]. Thus, there is a strong need for plant morphologists and biophilosophers to complement the research progress in developmental genetics of *Utricularia* and allies. There is a rich literature on comparative development and morphology in Lentibulariaceae provided by botanists dating back more than a hundred years (as summarized in [13,22]).

**Leaves, shoots, and stolons as descriptive terms for bladderworts.** Peter Taylor [212], in his marvelous book on bladderworts taxonomy (genus *Utricularia*), used structural terms such as leaves, stolons, and air shoots for the description of the vegetative bodies in bladderworts. He was well aware that a clear-cut and consistent terminology in plant morphology is needed to become understandable for a reading audience but may not be adequate to the reality. Taylor [212] (p. 6) wrote about plant architecture in bladderworts: "Quite apart from the traps themselves the organization of the vegetative organs of *Utricularia* is peculiar and quite different from that of other flowering plants... For taxonomic and descriptive purposes, whatever their true or theoretical nature, it is desirable to have a consistent terminology for the various organs. Such a terminology has evolved, through the work of Goebel, Lloyd, and others. Yet, because of their great diversity and often plastic nature, it is not always easy to apply the terms in a consistent manner, but for practical purposes, they are found to serve in the majority of cases." Taylor [212] (p. 8) continued about the strange *Utricularia* bauplan with a kind of British humor, while pointing to Rutishauser and Sattler [32]: "It seems reasonable to me that plants which have, in defiance of the general rule, no radicle or roots, may be allowed to have leaves which also disobey the rules. By calling them leaves a few difficulties do arise, but these are relatively unimportant in the context of the overall diversity and nonconforming nature of the vegetative morphology as a whole".

Species of *Utricularia* show various examples of **developmental mosaics** between structural categories typically referred to as leaf and shoot (see above). In certain aquatic members such as *U. purpurea*, the developmental pathway of the whole shoot is repeated within each compound leaf to an astonishing degree [199]. This, together with other observations, led Goebel [217] to the somewhat exaggerated conclusion that a primordium in *Utricularia* can grow into any organ such as a trap, leaf, green shoot (i.e., leafy stem), anchorage shoot, or inflorescence. This generalization is only partly accurate. The developmental and positional constraints in *Utricularia* deviate considerably from the rules of classical plant morphology. *Utricularia*'s plant body, thus, is better understood within the conceptual framework of continuum and process plant morphology [13,18,22]. Based on genomic data, the developmental geneticists Silva et al. [214] (p. 15) presented the following evolutionary hypothesis with respect to the two model species: "*Utricularia gibba* seems to have a more severe degree of fuzzy Arberian morphology, such as no clear delimitation of distinct vegetative organs. In contrast, *U. reniformis* presents a more traditional vegetative organ delimitation (as stems and leaves), similar to other angiosperms."

Members of the morphological school of Rolf Sattler [18,34,138,156] as well as Reut and Plachno [218] represented the vegetative bodies of aquatic and terrestrial bladderworts (e.g., *U. foliosa, U. dichotoma*) and other morphological misfits as combinations of developmental processes using

multivariate statistical analyses. Plant organs such as watershoots, leaves, or bracts in *U. foliosa*, also various kinds of stolons in *U. dichotoma* are identified in the morphospace as specific process combinations (see Sections 3.6 and 3.7). No doubt, the use of process combinations to describe plant structures makes communication among scientists difficult. Nevertheless, one of the great strengths of this approach is that categorical terms such as leaf and shoot serve only as placeholders for combinations of developmental processes that locate the organs in the morphospace. Gene expression patterns of model bladderworts (such as *U. vulgaris*, *U. gibba*) and related *Genlisea* species may finally be annotated to the morphospace by associating them with the combination of processes that are found in the parts where the genes are expressed [34,216,219,220].

How to avoid inconsistencies regarding the seeming lack of roots in *Utricularia?* Difficulties in distinguishing "root identity" and "shoot identity" (leafy stolons) are known from Lentibulariaceae, such as bladderworts (*Utricularia*) and butterworts (*Pinguicula*):

**Scenario A = 'Loss-of-Root' hypothesis:** Usually it is said that *Pinguicula* has roots, and the sister genera *Genlisea* and *Utricularia* lack them. Most botanists and developmental biologists insist that only the more basal genus *Pinguicula* still has true roots, whereas the members of the more derived *Genlisea–Utricularia* lineage have lost them completely [214,216,219–223]. Peter Taylor [212] (p. 6) came up with the following conclusion for *Utricularia*: "Roots are always absent, but organs that resemble and function as roots (here termed rhizoids) are usually present."

A main argument in favor of this scenario is the loss of various root-specific genes in *Utricularia gibba*, as summarized by Renner et al. [223]. According to them, various genes responsible for the growth of roots and root hairs are lacking (or apparently so) in *U. gibba*. Renner et al. [223] summarized (p. 147): "Taken together, loss of genes important for root and shoot morphogenesis may be correlated with the absence of an obvious root in *U. gibba* and could also help to define its unique bauplan suited to an aquatic habit."

**Scenario B = 'Root–Stolon Transformation' hypothesis:** *Utricularia* stolons may have arisen from what are called "roots" in *Pinguicula* just by adding exogenous leaves to the root surface, as already proposed by Brugger and Rutishauser [224]. This would explain the high degree of similarity of the stolons (stems) of various bladderworts with *Pinguicula* roots. Stolons of various terrestrial *Utricularia* species show an "awkward" phyllotaxis pattern (known as orthomonostichy) with all leaves arranged along a single stem sector [13,22,34,218]. According to the continuum plant morphology, the roots were not completely lost in the *Genlisea–Utricularia* lineage. The ancestral roots (as still present in *Pinguicula*) evolved exogenous green appendages that can be called "leaves" (an idea anticipated a hundred years ago by Arber [211]). Thus, the developmental pathways for roots and shoots were blended (amalgamated) to some degree, perhaps due to co-option of genes usually acting in stems and leaves but not in roots. Arguments in favor of this "root–stolon transformation" hypothesis are as follows: (i) Several *Pinguicula* have roots without caps (e.g., *P. moranensis*) [13,34]. (ii) Various Utricularias (e.g., *U. longifolia, U. livida, U. sandersonii*) have straight stolon tips which look (including anatomy) similar to cap-less root tips of *Pinguicula*. When *U. longifolia* is cultivated in a hanging pot, the root-like stolons breaking through the bottom show positive geotropism. (iii) Conversion of root meristems to shoot meristems are also known from other angiosperms such as *Nasturtium* (Brassicaceae) and *Neottia* (Orchidaceae), pointing to a morphological correspondence between root and shoot (as discussed by Guédès [225]).

Many developmental genes involved in the morphology of the *Genlisea–Utricularia* lineage were uncovered within the past few years. Genome analyses of *G. aurea, U. gibba*, and *U. vulgaris* (all of them apparently rootless) showed the presence of a considerable number of root-specific genes in their vegetative bodies [219,222,226,227]. For instance, genes involved in the root hair formation are present in *Utricularia*. Thus, the absence of a typical root can be a result of the expression (or lack thereof) of specific transcription factors and, therefore, not a result of lacking root developmental genes (Miranda, pers. comm.).

The 'loss-of-root" hypothesis (A) and the "root–stolon transformation" hypothesis (B) in the *Genlisea–Utricularia* lineage may merge into one when developmental processes and gene actions are emphasized instead of arbitrary structural categories. In the context of process morphology (getting rid of structural categories), both seemingly contradictory views are complementary perspectives of the same reality. The apparent loss of roots in *Utricularia* and *Genlisea* as compared to *Pinguicula* is a wonderful paradoxon where two contradictory perspectives express the same developmental reality. Both root and shoot meristems of vascular plants are regulated by common genetic mechanisms [223,228,229]. Reut and Plachno [218] (p. 17) hypothesized the following about stolon evolution in *U. dichotoma* as basal member of the genus *Utricularia* (subgenus *Polypompholyx*): "In terms of a continuum or morphocline of process combinations, and in the phylogenetic context, developmental processes for 'shoot' (e.g., branching) and developmental processes for 'leaf' (e.g., a determinate growth period) may have been added to a 'reduced *Pinguicula* root'." However, the *Utricularia* stolons should not be labelled as "obvious roots" or even "true roots". We should only keep in mind scenario B as complementary perspective: What is usually called "stolons" in *Utricularia* may have evolved from roots (cf. *Pinguicula*) that started to produce exogenous leaves.

Somewhat comparable to Arber's partial-shoot theory of the leaf (Sections 3.4 and 4.1), roots and shoots (stems, stolons) of vascular plants in general may be partially homologous, as summarized by Minelli [56] (p. 248): "From a phylogenetic perspective, the Continuum Model is consistent with the hypothesis that both leaves (e.g., [228,230]) and roots (e.g., [231,232]) are derived from shoot-like organs, at least in ferns and seed plants." Arber [86,103] already proposed the "partial-shoot theory of the root", coming close to the leaf-skin shoot model [97] (see Section 3.3), with the stem core being equivalent to the "root" and the stem cortex being formed by the elongated leaf bases, which cover the stem core like a skin. Arber [103] (p. 101) was aware of possible flaws of such a view when she wrote: "The supposition that the root may represent the 'core' of the shoot is not intended to imply anything so naively simple as that the root is anatomically a continuation of the central region of the shoot alone; this, indeed, would be obviously untrue, at least for the main root. It is held, rather, that the root is endowed with tendencies and capacities which correspond to those of the inner region of the shoot rather than those of its surface."

### 4.4. Flowering Plants—Flowers, Inflorescences, and Intermediates

The origin of flowers was a key innovation in the history of seed-plants between 150 and 190 million years ago. Typical flowers in flowering plants (angiosperms), especially in the large group of eudicots, consist of four whorls with sepals (usually green), petals (often showy and colorful), stamens (for pollen production), and carpels (often fused into one unit, the ovary). The latter contain the ovules which—once fertilized—become the seeds as result of sexual reproduction. The present case study does not allow us to consider all the wonderful morphological types of flowers and pollination syndromes in the c. 300,000 species of angiosperms, as presented in books and book chapters by, e.g., Endress, Johnson and Schiestl, Minelli, Ronse De Craene, Sattler, and Soltis et al. [56,233–237].

The developmental geneticists Elliott Meyerowitz and Enrico Coen (see e.g., [238]) proposed the ABC model of floral organ specification, mainly based on the two model plants Arabidopsis (thale cress, Brassicaceae) and Antirrhinum (snapdragon, Plantaginaceae). More recently, the ABC model turned into the more elaborated ABCE model. It explains the bauplan of typical flowers with four consecutive whorls (sepals, petals, stamens, carpels) by serial expression of organ identity genes, especially MADS box genes [124,235,239–243]. After having successfully explained the organ position in model organisms such as Antirrhinum and Arabidopsis, the developmental geneticists went on to explain flowers which form continua between sepals, petals, and stamens (in e.g., *Nymphaea*, waterlily). Thus, Soltis et al. [235,242] offered a **"fading borders" model** as a testable hypothesis for these floral organs of intergrading morphologies in basal lineages of flowering plants such as waterlilies, where the B-function genes are more broadly expressed. In the "fading borders" model, there is a gradual

transition of gene function across floral organs. This is unlike the classical ABCE model, where there is a sharp boundary between gene expression corresponding to floral organs [241,244–246].

Developmental mosaics between sepals, petals, and stamens are accepted by proponents of both classical and continuum plant morphology (see Section 3.2), because these kinds of floral appendages are understood—since Goethe—as modified leaves (phyllomes) [61,92,105]. Arber [28] (p. 55) summarized the history of this idea with emphasis on petals and stamens: "We may indeed agree with Goethe (1790) [92] and de Candolle (1827) [247] that petals and stamens show so much affinity that it is evidently reasonable to group them together. The petals will then be regarded as transition members between the vegetative and the actively reproductive parts of the floral shoot."

**Intermediates between flowers and inflorescences.** Still questionable in botany is the morphological significance of the angiosperm flower. Claßen-Bockhoff and Frankenhäuser [132] (p. 8) confessed: "While intermediate structures or even 'hybrids' between flowers and inflorescences are widely accepted (e.g., Prenner and Rudall [248]; Kirchoff et al. [34]), their morphogenetic and phylogenetic significance is not yet fully understood." Although many reproductive structures are easily identified as flowers, there are some examples lying in-between flowers and inflorescences, e.g., the "terminal flower-like structures", as observable in basal angiosperms and early monocots with elongate inflorescences [249].

Consider for example the staminate unit ("male flower") of the castor oil plant (*Ricinus communis*) that is known for its branched "staminal trees". Each unit may be seen as a single male flower with multiple branched stamen-fascicles [35,248,250,251]. Claßen-Bockhoff and Frankenhäuser [132] presented a somewhat contradicting view: Each "male flower" (branching staminate unit) of *Ricinus communis* is interpreted as a multi-flowered unit composed of highly reduced uni-staminate flowers, similar to what is known from *Euphorbia* within the same family as *Ricinus.*

Gene expression patterns are used to distinguish between flowers and inflorescences in Euphorbiaceae to which *Ricinus* belongs. Prenner et al. [251] found the expression of the LEAFY (LFY) protein not only in individual flower primordia of *Euphorbia*, but also in the primordium that leads to a group of flowers (called "cyathium"), indicating that inflorescence meristems may be genetically similarly regulated like floral meristems. Additional studies in various flowering plant families illustrate that one should be careful using gene expression patterns for floral organ homology [115,246,252].

Claßen-Bockhoff and Frankenhäuser [132] proposed floral unit meristems (FUMs) as intermediate stages between inflorescence and flower meristems: "Ongoing fractionation defining the transition from flowers to floral units may be based on a delay in the expression of floral organ identity genes." Claßen-Bockhoff and Frankenhäuser [132] remained vague with respect to male reproductive units (inflorescences, flowers?) in *Ricinus*. They concluded: "If the staminate unit is nevertheless interpreted as a flower (which is possible), one has to accept a completely new stamen structure and to disregard the meristem similarities between *R. communis* and floral units. Maybe, it is too early for a final morphological interpretation of the staminate units in *Ricinus*, as there is too little known about FUMs (=floral unit meristems), their diversity, and genetic regulation." FUMs allow a new view on intermediate stages/structures between flowers and inflorescences, because all types of reproductive meristems originate from shoot apical meristems. Somewhat similar fractionation steps as found in *Ricinus* reproductive development are observable inside the branching flowers of the tropical waterlily *Nymphaea prolifera,* leading to a multitude of sterile "daughter flowers" and "granddaughter flowers" that serve as vegetative propagules [34,253].

**Shoot-bearing leaves in *Hooded* barley.** Epiphyllous flowers are known from several angiosperms [95]. The developmental genetics behind this type of ectopic flower position is not yet fully understood. There is one nice example that was described and illustrated by Agnes Arber in her book on grasses [254]: the shoot-bearing leaf in Nepal barley (*Hordeum vulgare* var. *trifurcatum*). It shows epiphyllous spikelets arising from the awned bract (called "lemma" in grasses). By examining their relative positions, Arber [254] (p. 312) concluded that the lemma (bract) "which is a *leaf* member,

behaves to the accessory spikelet in all respects as if it were that spikelet's parent *axis* [her italics]." The very same situation is found in the *Hooded* mutant of *Hordeum vulgare* which was described later [255,256]: In *Hooded* barley plants, one or more extra flowers (spikelets) develops at the site of transition between bract ("lemma") and awn. Molecular studies [257–259] have elucidated that the *Hooded* phenotype of barley is caused by a duplication in a homeobox gene intron. Williams-Carrier et al. [259] suggested that the inverse polarity of the ectopic spikelets seen in the *Hooded* mutant of barley results from the homeotic transformation of the lemma awn into a reiterative inflorescence axis. This is an example of conversion of organ identity: a leaf part (the awn) is converted into a shoot axis. Neither Arber's book on grasses [254] nor the *Hooded* barley are mentioned in a recent review [260] on the developmental genetics of grass inflorescences.

### 4.5. Phyllotaxis: The Algorithmic Beauty of Plants

Shoot apical meristems (SAMs) in vascular plants give rise to organs at their flanks in a periodic and predictable pattern. Phyllotaxis is the spatial arrangement of lateral organs along a stem, within a rosette or shoot bud, or in a flower. The repeated lateral organs involved are usually leaves or flower parts. The study of phyllotaxis covers pattern formation in and around SAMs of vascular plants. Some phyllotactic patterns are easy to observe and characterize whole groups of plants. For example, leaf whorls with several units emerging from the same node are typical for horsetails (*Equisetum*), a group of flowerless and seedless vascular plants related to ferns [56,199]. Another group of regular phyllotactic patterns are the spiral ones, when the young leaves (or floral parts) arise one by one around the periphery of the SAM. For the past two hundred years, botanists and mathematicians tried to answer the question why most spiral patterns in seed plants approach the Fibonacci angle (c.137.5°), dividing the full circle (360°) according to the Golden section [168,261–267]. Barabé and Lacroix [268] summarized historical aspects and new research trends in a concise book entitled "Phyllotactic Patterns—a Multidisciplinary Approach". Phyllotaxis research applies mathematical models on pattern formation without bothering too much about the morphological significance of the plant organs involved [269]. In most cases, the lateral structures (appendages) arising at the periphery of SAMs are leaves, or axillary flowers, subtended by bracts in, e.g., the flower-heads of Asteraceae. There are only few cases known where leaves are replaced by lateral shoots or flowers at SAMs, e.g., in waterlilies (Nymphaeaceae) [253,270,271]. Natural examples of shoot or flower development from primordia occupying leaf sites are also known in clubmosses (*Huperzia*) [272,273], and *Utricularia* (see Section 4.3).

Process thinking and the continuum approach seem to be quite useful in phyllotaxis research. For example, structural units of seemingly unequal morphological significance (e.g., leaves, leaflets, and stipules) can be involved in the formation of leaf whorls of various flowering plants [35,199,200,266]. Leaf rosettes with attractive Fibonacci spiral patterns can even result from repeated sympodial branching (i.e., determinate shoot modules) combined with flowering, as found in *Pinguicula* [274]. Moreover, nature is not always as orderly as we think. There are many examples of vegetative and floral meristems where spirals and whorls are replaced by more irregular or even chaotic patterns, including clustered leaves on one side of the SAM. For example, irregular (chaotic) initiation of c. 200 stamens is observable in flowers of ylang-ylang (*Cananga odorata*, Annonaceae) [200,266,275].

Barabé and Lacroix [268] describe the interplay of biophysical parameters (e.g., tissue tensions in and around the SAM), size ratios between SAM and incipient leaves, and endogenous factors such as phytohormones (especially auxin) and gene expression that seem to be responsible for the emergence of regular patterns [261]. Molecular developmental geneticists already detected several genes that are involved in the emergence of disordered or perturbed patterns. Phyllotactic mutants, however, do not directly influence the size of the divergence angles (e.g., stronger deviations from the Fibonacci angle). They mainly change the three-dimensional shape and size of the SAM as the site of self-organizing processes [261,268,276,277]. During plant growth, including size increase of the SAM, various transitions between whorled, intermediate, and spiral patterns may occur, also depending

on the taxon involved [263,265,278]. Clubmosses (*Lycopodium* and allies), for example, switch more easily between patterns that are rare to very rare in seed plants [273,279,280]. Dynamic models and computer simulations based on process thinking, similar to those already presented for transitions in phyllotaxis, will help in the near future to better understand the complete architecture of vascular plants [155,168,268].

**Leaf initiation without an obvious SAM:** Finally, we should keep in mind that a shoot apical meristem (SAM) is not a "sine qua non" for leaf inception in flowering plants: (i) There are Australian *Acacia* spp. (e.g., *A. verticillata, A. baueri*) with "leaf whorls" that increase the number of whorl members by initiation of additional leaves (phyllodes) in meristematic stem zones below the proper SAM, as already described by the famous Wilhelm Hofmeister in 1868 [199,266,281]. (ii) The river-weeds Podostemaceae (adapted to river rapids and waterfalls in the tropics), especially its subfamily Podostemoideae, reveals leaf initiation from the adaxial side of an already formed young leaf. The shoot lacks a recognizable SAM with permanent stem cells. The SAM is cryptic or even lacking. A new leaf primordium in a vegetative shoot tip develops from the base of the opposing second youngest leaf primordium. The initiation of a new leaf primordium appears to be associated with degeneration of neighboring cells, as shown by Japanese botanists [82–84,282]. This kind of leaf formation without SAM is repeated, resulting in a chain of leaves or phytomeres (see Section 3.3).

## 5. Conclusions: Various Ways to Express Plant Growth and Architecture as Process Combinations and Developmental Continua

A sentence credited to the Greek philosopher Heraclitus of Ephesus (c. 535 BC–475 BC), "everything flows—nothing stands still", was used many times as a metaphor in lyrics and common speech. Heraclitus probably was one of the first process philosophers [38,39]. "Everything flows" is also the title of a book [33,40,283] as well as a paper [284] on processual philosophy of biology. The present essay focused on the philosophical foundations of process thinking and continuum views in biological sciences, as summarized by Jaeger and Monk [284]: "The process perspective holds that reality is fundamentally dynamic: the basic constituents of the universe are interconnected sequences of occurrences or events. Accordingly, reality must be described and explained using explicitly dynamic, that is processual, concepts rather than notions representing static 'things' or entities."

As a botanist with a passion for plant morphology (the main issue of the present essay), I have to admit that many interesting aspects of process philosophy could not be discussed here, including the heuristic value of continuum views in biological disciplines beyond plant morphology. For example, other biological disciplines such as systems biology, neurobiology, and cognitive sciences seem to be far ahead of plant biology/morphology with respect to process thinking. Jaeger and Monk [284] presented an example: "It is impossible to think about the nerve impulses that provide the substrate for thought, and ultimately the higher-level phenomena of mind and consciousness, in any other than dynamic terms." Also not covered by my essay are morphogenetic fields as dynamical modules and promising steps towards dynamical systems biology [16,17,285].

With respect to plant sciences, Sattler [6] (p. 61) admitted: "Although it seems easy to say that everything flows (changes), to arrive at a completely dynamic view of plant morphology is not an easy task." The present essay emphasizes the need for complementary perspectives, continuum, and process thinking in EvoDevo of both plants and animals. Historical roots of this open-minded way to look at growing organisms, especially plants, go back to Johann Wolfgang von Goethe, Agnes Arber, and Rolf Sattler. Multicellular plants such as seed plants, ferns, and bryophytes usually show algorithmic growth with continued branching and repetition of modules. The acceptance of growing plants as dynamic continua allows developmental geneticists and evolutionary biologists to move towards a more holistic understanding of plants in time and space [1,273].

There are several ways to understand the complexity, continuity, and dynamics of plant growth and architecture:

(i)    Recognize examples of unusual morphologies in plants by describing them as **morphological misfits** (see Section 3.2, Sections 4.1–4.4). Flowering plants with bauplans that deviate strongly from the approach of classical plant morphology were labelled as "morphological misfits" by Bell [75]. Being a misfit is not the problem of the plant, but the problem of our inadequate thinking and concepts. Morphological misfits do not fit classical plant morphology, which, however, is still useful as a rule of thumb in many usual (or normal) groups of flowering plants.

(ii)    Accept **scientific perspectivism** and complementary ways of describing the same plant in space and time: Stress that one kind of interpretation is often not enough to explain the architectural complexity. For example, the modular growth of a leafy shoot may be described by the phytomere model as well as by the traditional shoot–leaf model (see Section 3.3) [32].

(iii)    Consider the **principle of iteration** in plant development. A subunit such as a foliage leaf repeats to some degree the development of the whole shoot. This view coincides with the identity-in-parallel concept and the "partial-shoot theory" of Agnes Arber [28,49,72]. It has its counterpart in the anchor concept in zoology, where the paramorphism concept was proposed by Minelli [106,107] for multicellular animals such as tetrapods: Animal appendages can be regarded as a partial repetition of the main body axis (see Section 3.4).

(iv)    Use **process morphology** and mathematical tools to define intermediacy between typical plant organs: Process thinking and the continuum approach in plant morphology allow us to perceive and interpret growing plants as developmental continua, as process combinations rather than as assemblages of structural units ("organs"), such as roots, stems, leaves, and flowers. Therefore, we may use strict sets of developmental processes for defining leaves vs. stems vs. roots. Then let us code these characteristics and use statistical tools like principal component analysis (PCA) for the distinction of typical plant organs (as found in many vascular plants) and developmental mosaics between structural categories. Homology includes partial homology and quantitative homology, as proposed by Sattler [6,113,119,133]. In vascular plants, this leads to a continuum between structural categories (plant organs) such as roots, stems, leaves, and even multicellular hairs/trichomes. How intermediates between typical plant organs are best described has been discussed by Kirchoff et al. [34] and Lacroix et al. [49]. For example, the structures observed in the bladderworts (*Utricularia*) may be called leaf and stem for convenience, as done by Taylor [212], although other authors [22,218] interpreted them as developmental mosaics including root components (see Section 4.3).

(v)    Accept **developmental genetics** as the 4th homology criterion for defining the morphological significance of unusual plant structures. Traditionally, three homology criteria were used: position, special quality, and the existence of intermediates [62,108,113,117,118]. Unlike the more holistic Sattler school, reductionistic biologists such as Scheres et al. [286] (p. 963) emphasized the primacy of molecular genetics over traditional morphology/anatomy: "Regardless of how much faith one has in anatomical definitions, they should not be taken as more than a means of communication prior to subsequent genetic analysis." Similarly, developmental geneticists may insist on the primacy of organ-identifying genes over the three traditional homology criteria [125]. For example, by stating that the bladderworts (*Utricularia*) lack important genes for roots, there is a genetic basis for the lack of (typical) roots in the bladderworts [223] (see Section 4.3).

(vi)    Accept **developmental processes** such as homeosis, ectopic expression, blurring, and upgrading of organ identities in plant structures that transcend typical bauplans. Although the concept of homeosis is much older than developmental and molecular genetics, it gained much additional explanatory power with the discovery of homeotic genes (organ-identity genes), such as the MADS box genes explaining the bauplan of typical flowers in angiosperms [119,124] (see Section 4.4). When classical bauplan rules of vascular plants (consisting of roots, stems, and leaves) are violated, then it becomes difficult to clearly define and discriminate between the three types of organs. Rutishauser et al. [35] described several examples of plants "having identity crises".

Identity crises result from our inadequate vocabularies while describing and interpreting plant architecture in space and time.

(vii) Design **virtual plants** using iterated developmental processes. The development of new mathematical concepts and computational techniques for the description of growing plant structures can be based on developmental rules such as branching, repetition of growth units (e.g., phytomeres), and environmental parameters, as already done by Prusinkiewicz and colleagues [155,168,267,276,287].

(viii) Maybe we need to do more of what Agnes Arber and other morphologists since Goethe have done: Rely more on the visual abilities that draw people to biology. Get a **feeling for the organism** [8–12,61,134,288,289].

(ix) Last but not least: **Celebrate** your achievements towards process and continuum thinking with a sip of Agnes-Arber Gin: "A fantastic gin celebrating the renowned botanical historian Agnes Arber". **Cheers!** https://agnesarbergin.signature-brands.co.uk/online/—In addition, you may listen to the song on EvoDevo (Despacito Biology Parody), a music video performed by the Canadian science communicator and youtuber Tim Blais (A Capella Science): https://www.youtube.com/watch?v=ydqReeTV_vketlist=PL20YbtNRgutzZftYyTI_p2G3ttd4R9dVs.

**Funding:** This research received no external funding.

**Acknowledgments:** I am grateful to Alessandro Minelli for inviting me to contribute to the special issue on "Renegotiating Disciplinary Fields in the Life Sciences" and for commenting on a draft version of this contribution. I would also like to thank Denis Barabé, Volker Bittrich, Christian Lacroix, Rolf Sattler, and the two anonymous peer reviewers for their criticisms, comments, and suggestions.

**Conflicts of Interest:** The author declares no conflict of interest.

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
