# Peer review of "EvoDevo: Past and Future of Continuum and Process Plant Morphology"

_philosophies, doi:10.3390/philosophies5040041_

Round 1

Reviewer 1 Report

I had a great deal of difficulty understanding most of this manuscript. It is quite long, and there does not seem to be a central point that the author expects the reader to understand. Certainly, the author’s knowledge of the literature on process morphology is encyclopedic, and he demonstrates this knowledge in all parts of the manuscript. However, other than the fact that the author thinks that process morphology and the related continuum morphology are important for understanding plant structure, I do not know what he hopes me to learn. Is there a central point that I have missed? I have tried reading the opening few pages several times, and do not find a unifying theme. I had hoped that a central theme would emerge as I continued reading, but I was unable to identify a central message in the first several sections. The section on case studies, Section 4, is more coherent and it might be possible to enlarge this section to help clarify the author’s points. The opening part of the Conclusion, Section 5, only expresses the authors regret that he could not cover all aspects of process philosophy in this long paper. Certainly, every author who has struggled with a complex subject can sympathize with him in this frustration, but these paragraphs do not help the reader learn anything from the manuscript. The summary points (i – viii) in Section 5 are more helpful, and with the case studies from Section 4 could be expanded into a useful paper. To do this, much of Sections 2 and 3 would have to be removed. I found these sections very hard to understand and did not feel that I learned anything from multiple readings.

Let me give an example of some of the problems I had in the first sections. Lines 733 to 744 are about gene expression in higher plants, specifically gene expression in the egg. This is part of section 3.8 (“Possible lack of one-to-one correspondence between structural categories and gene expression in higher plants”) of the manuscript. When we look at the citation (citation 151) for the main quote on these lines, we find it is from a book review of a book that is primarily about animals. What is one to understand from this? Why is a book review cited and quoted as a major reference? Reading on in the rest of section 3.8 we come to line 745, which is on a completely different subject, plant structure ontology. What is the relationship between plant structure ontology and gene expression in the egg? I am at a loss to understand this relationship. The next sub-section, beginning on line 786 is about homology. All of these sub-sections, gene expression, plant ontology, homology must be related somehow because they are all part of section 3.8. I do not, however, understand their relationship. I do not understand how these three subjects all relate to the possibility that there may not be a one-to-one correspondence between structural categories and gene expression.

There are, unfortunately, many more examples like this throughout Section 2 and 3 of the manuscript. Perhaps if I had the same depth of knowledge as the author I would see the connections, but I am not familiar with every manuscript that he cites, and I am at a loss to see the connections between many of the points that he raises.

Section 4, Case Studies, is better. Process and fuzzy morphology do seem to be important for understanding the examples used in the case studies. I found this section to be easier to understand. The examples provide good evidence for the use of process and fuzzy morphology. I think the manuscript would be strengthened if the author removed Sections 2 and 3 of the manuscript and moved some of the material from those sections to the discussion of these case studies. He would have to be careful not to overwhelm the descriptions of plant morphology, but some of the material from earlier in the manuscript could be profitably used here. Much of it would however have to be discarded. I found the manuscript to be too long and too obscure in its present form.

-----------

Here are some further examples of the difficulties that I had with the first two sections of the manuscript.

In the following, L = Line.

L97-102: I am unclear how the section on individuality relates to the ideas of perspectivism. Section 2.2 needs a bit of tightening up to clarify the authors’ meaning.

Section 2.3: I am not sure that there is such a clear-cut difference between what the author calls conceptual realism and what he refers to as perspectivism. If we think about the process of biological classification, we could say that every species name represents one perspective on the reality that exists in nature. Each time a species is named a specific taxonomy is constructed according to the ideas of the author who applies that name. A different taxonomist looking at the same data will all too often come to a different circumscription and therefore apply a different name. These names are categorical, nonoverlapping (within a given frame of reference = within a specific taxonomic work) and yet are great examples of perspectivism.

L119-120: I do not understand the transition between the discussion of conceptual realism on line 119, and the discussion of biological individuality on line 120. Is the author suggesting that biological individuals are in some way “conceptually real” whatever that means? I really do not understand the author’s points on these lines.

L124: Terms like “heap” are mass nouns. A mass noun is a word that refers to a group of non-countable “things.” Words like advice, bread, and knowledge are mass nouns. There has been a good deal of work in cognitive psychology to try and understand how people treat mass nouns. I am not sure that it is fair to say that mass nouns do not have clear sets of properties. The author would need to do a good deal more work to understand and present the research on mass nouns before he can make this claim.

L149-154: the fact that Mayr was not influenced by Whitehead really does not say very much about the full field of evolutionary biology, especially about the work that has been done since Mayr’s death. The author’s main thesis seems to be that perspectivism is needed to understand biological processes, but he makes his arguments from an essentialist point of view. He does not speak about different authors perspectives of evolutionary biology. He chooses one, essential, evolutionary biologist and criticizes his approach. Perhaps the author would consider taking a less essentialist approach when making his arguments.

L155-162: I think that one can make a strong argument that the whole field of systems biology has adopted the type of process approach that the author advocates. I suggest that he investigate this field and included discussion of it in the manuscript. I did see that he mentions systems biology in the final paragraphs of Section 5.

L163-174: I did not understand what the author hopes me to take away from this paragraph. The beginning sentence is so long and has so many phrases that I was unable to clearly understand its meaning.

L179-190: I have two comments on the section. First, it would be good to have some examples of disciplines in which the transformation of structures (not ontogenies) is discussed. I believe that there are many phylogenetic analyses that could serve this purpose. Second, it is never been clear to me how we should describe the evolution of ontogenies. Stephen J Gould certainly attempted this in his work on heterochrony, but I am unaware of anyone who has attempted it on a phylogenetic basis. This is probably just my ignorance, and it would be good if the author could find some examples and include them in the section. I think that is necessary to have examples of the type of analyses he would like to see.

L209: I do not understand how this section follows from the previous sections. That is, I do not understand their connection.

L241: Once again, I do not understand how this section relates to the previous ones. It seems like each section presents its own ideas, which are unrelated to what has been said before. I also do not understand the structure of this paragraph. The author begins and ends by praising the SPAC, but in the middle of the paragraph he criticizes it. This leaves me confused as to what he hopes me to learn from this section.

L259: An organism’s morphology is part of its phenotype and is therefore the basis of selection. Morphology is not only important for systematics.

L265: the term bauplan needs to be defined.

L278-292: the information in this paragraph is certainly interesting, but I do not understand how it relates to the author’s main point.

L439: I was unable to understand the phytomere model from the description that is presented here.

L452-453: these arguments may have taken place in the past, but they are not part of contemporary botanical research.

L454-455: this sentence seems to be the heart of the author’s argument. “We can overcome this attitude by the acceptance of perspectivism and complementarity.” It would help me greatly if he would make this point at the beginning of the manuscript, and then expand on it with examples like those in Section 4. I believe that most of Sections 2 and 3 could be removed.

L464: I do not understand how this section relates to the previous one, or to the main point of the article. The author has already pointed out that Sattler’s theory was first published 50 years ago. I am not sure why it is important to repeat that fact here.

L489-500: this section began with the discussion of Sattler but quickly changed to discussion of Arber. I did not understand this transition, other than the fact that Arber’s views were an influence on Sattler.

L528: The method suggested by Jaramillo & Kramer for investigating the homology of morphological structures is certainly interesting, but I do not understand why it is reproduced here. What does the author hope that I will understand from a list of these four processes?

L541-562: I do not understand the point of this paragraph. Is the author arguing against Jaramillo & Kramer’s processes? I do not understand the relationship between this paragraph, the preceding paragraph, and the following paragraph. What is it that the author hopes for me to understand?

L582-585: but it is also possible that process morphology has not received more attention because few or no pressing scientific questions demand a process approach. The one exception might be systems biology, which is very process oriented, but which had not been mentioned up to this point. The author seems to want to claim that process morphology, partial homology, etc. have not received sufficient attention in the biological community, while ignoring those areas where they have received attention.

L615: are not developmental routines also categories? How does it help to switch one type of category for another? Is the author suggesting that we replace structural categories with process categories, and then create combinations of these process categories? Do not both of these processes imply some degree of essentialism? I am not sure that we can ever avoid some degree of essentialism.

L633 (and prior mentions of bladderworts): It would help me understand the author’s points if he would provide a description of why bladderworts are morphological misfits.

L645-646: is the author suggesting that Zimmerman’s telome theory is an example of Fuzzy Arberian Morphology? I have always understood Zimmerman’s approach to be quite typological. He does talk about the gradual evolution of leaves, but he envisions this taking place by the rather rigid transformation of one structural category into another. I can see that Zimmerman does speak about the processes that are involved in these transformations, but his processes are purely hypothetical (and also are categorical). The Zimmerman processes have never been shown, to my knowledge, to actually exist in nature. They seem to me to be ideal transformations of one typological form into another. I do not understand how Zimmerman’s work is a good example of process morphology.

L674-675: I do not understand this sentence. “From a developmental perspective, canonical plant shoots (as generally recognized in vascular plants) can be defined as a process, i.e., they develop iteratively from an apex to produce lateral organs.” I do not understand what it means for a shoot to develop iteratively. Also, are shoots not process combinations? I do not understand how a shoot can be equated with a single process.

L675-676: The protonema of a moss does not have an axial body plan. The above ground stems of the mature gametophyte do. I do not think that there is any controversy about the fact that the mature gametophytes are at least shoot-like. I am not sure why the author thinks this is an important point.

L671-681: I do not understand what the author wants me to know after reading this paragraph. He begins by talking about process morphology and ends by talking about the structure of river-weeds shoots, but he does not relate these structures to process morphology.

L690-704: I think that the author makes some good points in the first part of this paragraph, but I feel that these points are diluted by the long quote at the end of the paragraph. The two parts of the paragraph do not seem closely related to me.

Author Response

I had a great deal of difficulty understanding most of this manuscript. It is quite long, and there does not seem to be a central point that the author expects the reader to understand. Certainly, the author’s knowledge of the literature on process morphology is encyclopedic, and he demonstrates this knowledge in all parts of the manuscript. However, other than the fact that the author thinks that process morphology and the related continuum morphology are important for understanding plant structure, I do not know what he hopes me to learn. Is there a central point that I have missed? I have tried reading the opening few pages several times, and do not find a unifying theme. I had hoped that a central theme would emerge as I continued reading, but I was unable to identify a central message in the first several sections.

My response: Referee 1 (“Ref1”) got my main message “Process morphology and the related continuum morphology are important for understanding plant structure”. I really tried to introduce this theme because many biologists – including Ref1 – still have difficulties to think in continua and processes.

The section on case studies, Section 4, is more coherent and it might be possible to enlarge this section to help clarify the author’s points. The opening part of the Conclusion, Section 5, only expresses the authors regret that he could not cover all aspects of process philosophy in this long paper. Certainly, every author who has struggled with a complex subject can sympathize with him in this frustration, but these paragraphs do not help the reader learn anything from the manuscript.

Opening part of the Conclusion, Section 5, I did not omit because Jaeger & Monk {274} {275} are leading proponents of systems biology (a discipline seemingly overlooked in my manuscript, at least according to Ref1): As a botanist with a passion for plant morphology (= the main issue of the present essay), I have to admit that many interesting aspects of process philosophy could not be discussed here, also not the heuristic value of continuum views in biological disciplines beyond plant morphology. For example, other biological disciplines such as systems biology, neurobiology and cognitive sciences seem to be far ahead of plant biology/morphology with respect to process thinking! Jaeger & Monk {274} bring an example: “It is impossible to think about the nerve impulses that provide the substrate for thought, and ultimately the higher-level phenomena of mind and consciousness, in any other than dynamic terms.” 

The summary points (i – viii) in Section 5 are more helpful, and with the case studies from Section 4 could be expanded into a useful paper. To do this, much of Sections 2 and 3 would have to be removed. I found these sections very hard to understand and did not feel that I learned anything from multiple readings.

I want to justify the need of the sections 2 and 3 as parts of the revised manuscript version. For a philosophical journal a philosophical introduction as I presented it in sections 2 and 3, especially in section 2, is required and necessary.

Let me give an example of some of the problems I had in the first sections. Lines 733 to 744 are about gene expression in higher plants, specifically gene expression in the egg. This is part of section 3.8 (“Possible lack of one-to-one correspondence between structural categories and gene expression in higher plants”) of the manuscript. When we look at the citation (citation 151) for the main quote on these lines, we find it is from a book review of a book that is primarily about animals. What is one to understand from this? Why is a book review cited and quoted as a major reference? Reading on in the rest of section 3.8 we come to line 745, which is on a completely different subject, plant structure ontology.

What is the relationship between plant structure ontology and gene expression in the egg? I am at a loss to understand this relationship.

My response: The Plant Structure Ontology (PSO: www.plantontology.org) is thought to be a controlled vocabulary of botanical terms describing morphological and anatomical structures representing organ, tissue and cell types, and their hierarchical relationships. As explained by Ilic & al. (2007 in Plant Physiology) and repeated in the revised version of my PHILOSOPHIES manuscript: The PSO was developed by the Plant Ontology Consortium in response to the rapid proliferation of molecular sequences and databases, which has created data access problems for biologists. The main intent was to create a unified and hierarchical vocabulary of plant morphology and anatomy that can be used to describe spatial and temporal aspects of gene expression, because – traditionally - the same terms are sometimes applied to different plant structures in different taxonomic groups (see more details in Kirchoff & al. 2008 in Zootaxa).

The next sub-section, beginning on line 786 is about homology. All of these sub-sections, gene expression, plant ontology, homology must be related somehow because they are all part of section 3.8. I do not, however, understand their relationship. I do not understand how these three subjects all relate to the possibility that there may not be a one-to-one correspondence between structural categories and gene expression.

There are, unfortunately, many more examples like this throughout Section 2 and 3 of the manuscript. Perhaps if I had the same depth of knowledge as the author I would see the connections, but I am not familiar with every manuscript that he cites, and I am at a loss to see the connections between many of the points that he raises.

I agree: Some parts of my paper (including the revised manuscript) need a lot of background knowledge to be fully understood.

Section 4, Case Studies, is better. Process and fuzzy morphology do seem to be important for understanding the examples used in the case studies. I found this section to be easier to understand. The examples provide good evidence for the use of process and fuzzy morphology. I think the manuscript would be strengthened if the author removed Sections 2 and 3 of the manuscript and moved some of the material from those sections to the discussion of these case studies. He would have to be careful not to overwhelm the descriptions of plant morphology, but some of the material from earlier in the manuscript could be profitably used here. Much of it would however have to be discarded. I found the manuscript to be too long and too obscure in its present form.

I don’t omit the sections 2 and 3 as parts of the revised manuscript version. For a philosophical journal a philosophical introduction as I presented it in sections 2 and 3, especially in section 2, is required and necessary.

I am happy to realize that Ref1 accepts the heuristic value of process and fuzzy morphology for better understanding the examples used in the case studies (Section 4).

I tried to shorten some sections (see revised ms).

-----------

Here are some further examples of the difficulties that I had with the first two sections of the manuscript.

In the following, L = Line.

L97-102: I am unclear how the section on individuality relates to the ideas of perspectivism. Section 2.2 needs a bit of tightening up to clarify the authors’ meaning.

O.k. I agree to some degree. However, I did not omit the paragraph on the question “What is an individual in biology?" 

Section 2.3: I am not sure that there is such a clear-cut difference between what the author calls conceptual realism and what he refers to as perspectivism. If we think about the process of biological classification, we could say that every species name represents one perspective on the reality that exists in nature. Each time a species is named a specific taxonomy is constructed according to the ideas of the author who applies that name. A different taxonomist looking at the same data will all too often come to a different circumscription and therefore apply a different name. These names are categorical, nonoverlapping (within a given frame of reference = within a specific taxonomic work) and yet are great examples of perspectivism.

I agree with Ref1. The various ways to define “species” in biology are nice examples of perspectivism. However, this topic is beyond the themes I want to discuss in my review on “Past and Future of Continuum & Process Plant Morphology”. A nice contribution to perspectivism was provided by Igor Pavlinov (2020) in his PHILOSOPHIES article added to Minelli (ed) Renegotiating Disciplinary Fields in the Life Sciences” (see PDF under: https://www.mdpi.com/2409-9287/5/2/7). 

L119-120: I do not understand the transition between the discussion of conceptual realism on line 119, and the discussion of biological individuality on line 120. Is the author suggesting that biological individuals are in some way “conceptually real” whatever that means? I really do not understand the author’s points on these lines.

L124: Terms like “heap” are mass nouns. A mass noun is a word that refers to a group of non-countable “things.” Words like advice, bread, and knowledge are mass nouns. There has been a good deal of work in cognitive psychology to try and understand how people treat mass nouns. I am not sure that it is fair to say that mass nouns do not have clear sets of properties. The author would need to do a good deal more work to understand and present the research on mass nouns before he can make this claim.

The referee is criticizing the behavioral biologist Bernhard Hassenstein (1922-2016), not me! See more on him and his extraordinary wide philosophical horizon in https://de.wikipedia.org/wiki/Bernhard_Hassenstein . Already Hassenstein (1978) pointed out that “life” (also mass nouns such as “heap” and “bread”) must be seen as injunctions, i.e., as concepts that cannot be defined by a clear-cut set of properties (in Germanein Mittel der Begriffsbestimmung in Gegenstandsbereichen, in denen die Anwendung einer Definition nicht sachgerecht ist”). Illustrating the problem, Hassenstein asked the question “How many grains result in a heap?” Actually, the definition of taxa in biology such as “species” coincides with the “heap” problem. That means, many biological concepts may be “mass nouns” in the sense of Ref1.

L149-154: the fact that Mayr was not influenced by Whitehead really does not say very much about the full field of evolutionary biology, especially about the work that has been done since Mayr’s death. The author’s main thesis seems to be that perspectivism is needed to understand biological processes, but he makes his arguments from an essentialist point of view. He does not speak about different authors perspectives of evolutionary biology. He chooses one, essential, evolutionary biologist and criticizes his approach. Perhaps the author would consider taking a less essentialist approach when making his arguments.

My response to this criticism: It was not my intention to enter the field of evolutionary biology since Ernst Mayr although EVO-DEVO is nowadays an important discipline in order to combine evolutionary biology with developmental genetics. With my quotation on how Ernst Mayr thought about A.N. Whitehead (1861–1947), one of the founders of “process philosophy”, I just wanted to say that this kind of process thinking was not fashionable in developmental and evolutionary biology four decades ago.when Mayr wrote his book “The Growth of Biological Thought”. It is interesting to realize that Mayr (1982, 1984) nor Ilse Jahn (1982---2000) did also not mention the developmental biologist Wardlaw (1965, p. 371) who wrote “Organization is a continuum in the physical world. Organization is also a continuum in the ontogenesis and reproduction of the individual organism and in the phyletic line of which it is a component.”

L155-162: I think that one can make a strong argument that the whole field of systems biology has adopted the type of process approach that the author advocates. I suggest that he investigate this field and included discussion of it in the manuscript. I did see that he mentions systems biology in the final paragraphs of Section 5.

My comment: I agree with Ref1. Systems biology provides further examples of process thinking. However, this topic is beyond the themes I wanted to discuss in my PHILOSOPHIES review on “Past and Future of Continuum & Process Plant Morphology”. Ref1, however, should be aware that I have cited important systems biologists such as J. Jaeger, N. Monk, J. Baedke, M. Benitez & al. already in my first manuscript version. “Systems biology” combines quantitative experimental measurements with dynamical systems modeling, as described by Johannes Jaeger 2017 in “The importance of being dynamic: Systems biology beyond the hairball”, published as chapter in Philosophy of Systems Biology. Perspectives from Scientists and Philosophers (ed. S. Green). Actually, “systems biology” is not well-defined, as expressed by Amato in his PHILOSOPHIES review on “EvoDevo: An ongoing revolution?” [Philosophies 2020, 5(4), 35. https://doi.org/10.3390/philosophies5040035]. Amato (p.7 pf 21) wrote: “Systems Biology is an emerging approach of which it is difficult to provide a comprehensive definition; within this field are present various traditions, such as a physiological and a computational one. Anyway, to provide a general definition, systems biology “is the study of the behavior of complex biological organization and processes in terms of the molecular constituents” (Kirschner 2005, p. 504).”

L163-174: I did not understand what the author hopes me to take away from this paragraph. The beginning sentence is so long and has so many phrases that I was unable to clearly understand its meaning.

I shortened slightly the sentence on Baedke & Mc Manus as follows: “Baedke & Mc Manus (2018) have shown that not only structures but also developmental processes in organisms can be viewed as part of time scale hierarchies (labelled as ‘dynamic hierarchies’), including processual hierarchies between genotype and phenotype.” etcetc

L179-190: I have two comments on the section. First, it would be good to have some examples of disciplines in which the transformation of structures (not ontogenies) is discussed. I believe that there are many phylogenetic analyses that could serve this purpose. Second, it is never been clear to me how we should describe the evolution of ontogenies. Stephen J Gould certainly attempted this in his work on heterochrony, but I am unaware of anyone who has attempted it on a phylogenetic basis. This is probably just my ignorance, and it would be good if the author could find some examples and include them in the section. I think that is necessary to have examples of the type of analyses he would like to see.

Good idea! However, the discussion of phylogenetic analyses in combination with the transformations of structures / morphological characters (i.e., “not ontogenies”) is beyond the scope of the present manuscript.

L209: I do not understand how this section follows from the previous sections. That is, I do not understand their connection.

L241: Once again, I do not understand how this section relates to the previous ones. It seems like each section presents its own ideas, which are unrelated to what has been said before. I also do not understand the structure of this paragraph. The author begins and ends by praising the SPAC, but in the middle of the paragraph he criticizes it. This leaves me confused as to what he hopes me to learn from this section.

O.k. Paragraph on SPAC omitted in revised manuscript!

L259: An organism’s morphology is part of its phenotype and is therefore the basis of selection. Morphology is not only important for systematics.

I agree!

L265: the term bauplan needs to be defined.

I wrote (ms prior to revision lines 264-268): “Unlike most multicellular animals the plants usually have an open bauplan, with continued branching and repetition of subunits (plant organs) such as leaves, stems, roots, and flowers. This is especially true for seed plants whereas ferns (as another group of vascular plants) lack flowers and seeds, while mosses (bryophytes) lack also roots and may even lack differentiation into leaves and stems.” - It seems clear to me within this context that a bauplan (= body plan) is a set of architectural rules for the body (phenotype) of plants and other groups of organisms.

L278-292: the information in this paragraph is certainly interesting, but I do not understand how it relates to the author’s main point.

If leaves evolved several times and independently in the various groups of vascular plants, it will be questionable to homologize the leaves in general…

L439: I was unable to understand the phytomere model from the description that is presented here.

I improved the sentence in the revised ms: “In the phytomeric model the stem zone, called ‘phytomere’ (also written ‘phytomer’) consists of a leaf, its axillary bud, the node and one internode below the leaf.” – In the old ms version, the phytomeres were described and defined in lines 424 to 434 (ms prior to revision): “In the phytomeric model (also called “metameric” one) the stem zone, called ‘phytomere’ (also written ‘phytomer’) consists of one internode and the node where the leaf (or leaf whorl) is inserted.”

L452-453: these arguments may have taken place in the past, but they are not part of contemporary botanical research.

Not true! There are several developmental botanists who adopted – complementing the classical leaf-and-stem concept – the “phytomere” concept for better understanding the development of shoots in vascular plants. See White (1979, 1984), Müller & al. (2015, in Plant Signaling & Behavior) for Arabidopsis; Kinoshita & Tsukaya (2019, in Dev. Growth Differ.) for “One-leaf plants in the Gesneriaceae”.

L454-455: this sentence seems to be the heart of the author’s argument. “We can overcome this attitude by the acceptance of perspectivism and complementarity.” It would help me greatly if he would make this point at the beginning of the manuscript, and then expand on it with examples like those in Section 4. I believe that most of Sections 2 and 3 could be removed.

Line 450-451 of ms prior to revision: “We can overcome this attitude by the acceptance of perspectivism and complementarity (see paragraph 2.2).” – I made this point already at the beginning of the ms, i.e. in paragraph 2.2!!!

As already emphasized above, the sections 2 and 3 are needed as parts of the revised manuscript version. For a philosophical journal a philosophical introduction as I presented it in sections 2 and 3, especially in section 2, is required and necessary.

L464: I do not understand how this section relates to the previous one, or to the main point of the article. The author has already pointed out that Sattler’s theory was first published 50 years ago. I am not sure why it is important to repeat that fact here.

One sentence removed! However, Sattler’s fifty-year jubilee was mentioned only shortly as part of Abstract and Introduction.

L489-500: this section began with the discussion of Sattler but quickly changed to discussion of Arber. I did not understand this transition, other than the fact that Arber’s views were an influence on Sattler.

The thinking of Agnes Arber and Rolf Sattler are closely related…

L528: The method suggested by Jaramillo & Kramer for investigating the homology of morphological structures is certainly interesting, but I do not understand why it is reproduced here. What does the author hope that I will understand from a list of these four processes?

The four-steps recipe offered by Jaramillo & Kramer (2007) is a valuable procedure for developmental geneticists when they - tentatively – try to accept a 1:1 correspondence between morphological structures and gene functions.

L541-562: I do not understand the point of this paragraph. Is the author arguing against Jaramillo & Kramer’s processes? I do not understand the relationship between this paragraph, the preceding paragraph, and the following paragraph. What is it that the author hopes for me to understand?

The author’s hope is that the reader gets some information of the close vicinity of the concepts of “organ identity” (“meristem identity”) and “structural homology”. Partial homology as proposed by Sattler already 50 years ago tries to overcome shortcomings when plants are having identity crises, as illustrated with many examples by Rutishauser & al. (2008) in a chapter in the book “Evolving pathways. Key themes in evolutionary developmental biology“(eds. Minelli & Fusco). As mentioned above, I have nothing against the four-steps recipe offered by Jaramillo & Kramer (2007), when developmental geneticists tentatively accept a 1:1 correspondence between morphological structures and gene functions.

L582-585: but it is also possible that process morphology has not received more attention because few or no pressing scientific questions demand a process approach.

Ref1 does not seem to be aware of the beginning paradigm shift that is observable in the heads of many evolutionary and developmental biologists (including systems biologists) nowadays. Examples are presented below:

Biology in general: Systems biology as understood by Johannes Jaeger (2017, 2019; see also Jaeger & Monk 2015) strongly overlaps with process-oriented biology. About the “evolution of developmental systems” in e.g. dipteran insects Johannes Jaeger (2017, p.138) wrote: “I see two main points of intersection between systems biology as a general approach and the philosophy of science. The first concerns process philosophy, the second scientific perspectivism. Process philosophy argues that processes are more fundamental than substance (static things or entities) (Rescher 1996). In its most pragmatic form, process philosophy states that it is useful and important to study nature in processual terms. In other words, while it is important to know what a system is made of, it is the interactions between its components that define a system’s behavior (its dynamical repertoire) and its potential for future change. Things are only of interest as long as they affect other things. Inert entities are irrelevant. Thus, things can only be studied as parts of processes, which is exactly what systems biology is supposed to do. For this reason, process philosophy provides the proper epistemological foundation for the kind of systems biology I am talking about.”

Botany: I want to mention the new publication by Nikolov & al. (2019, in Current Topics in Dev. Biol. 131) on leaf development and evolution. They wrote (p.124): “It has been speculated that such convergences [between leaves and shoots] reflect the deep evolutionary origin of leaves from branched shoots, for example, by Arber (1950, reprint 2012), who emphasized the developmental similarities of shoots, leaves, and leaflets. Molecular evidence now helps make those debates more concrete and, in this section, we focus on cases where genetic evidence has provided insights into the causal molecular changes underlying diversity.”  --

Moreover the computational biologists Prusinkiewicz & Runions (2012) gave credit to Agnes Arber while simulating plant growth: “This similarity is consistent with the ‘partial shoot theory’ (Arber 1950), which emphasizes parallels between the growth of shoots and leaves. The strikingly different appearance of spiral phyllotactic patterns and leaves would thus result not from fundamentally different morphogenetic processes, but from different geometries on which they operate: an approximately paraboloid SAM dynamically maintaining its form vs. a flattening leaf that changes its shape and size.”

The one exception might be systems biology, which is very process oriented, but which had not been mentioned up to this point. The author seems to want to claim that process morphology, partial homology, etc. have not received sufficient attention in the biological community, while ignoring those areas where they have received attention.

As already mentioned above, systems biology provides further examples of process thinking. It is not true that in my manuscript systems biology is not included. Ref1 should be aware that I have cited important systems biologists such as J. Jaeger, N. Monk, J. Baedke, M. Benitez & al. already in my first manuscript version.

L615: are not developmental routines also categories? How does it help to switch one type of category for another? Is the author suggesting that we replace structural categories with process categories, and then create combinations of these process categories? Do not both of these processes imply some degree of essentialism? I am not sure that we can ever avoid some degree of essentialism.

I agree! A moderate essentialism cannot be avoided at all in biology! There are two extremes complementing each other: structure ontology (i.e. moderate essentialism with respect to nouns / structures) and process ontology (i.e. essentialism with respect to developmental or even evolutionary processes & mechanisms), as mentioned in my manuscript (version prior to revision on lines 162-169, see my comments above, citing Baedke & Mc Manus 2018).

L633 (and prior mentions of bladderworts): It would help me understand the author’s points if he would provide a description of why bladderworts are morphological misfits.

Fuzzy Arberian Morphology versus Classical Plant Morphology are described with respect to bladderworts (Utricularia) and allies by Rutishauser (2001 in Ann.Bot.), Rutishauser (2016 in Ann.Bot.). The present PHILOSOPHIES review does not allow to repeat all the details why bladderworts can be viewed as “morphological misfits”. However, as part of the case study 4.3. “Flowering plants – Bladderworts & allies (Lentibulariaceae)” several aspects (including complementary views on the seeming lack of roots in Utricularia) are described (see lines 1020 – 1169 in version prior to revision). 

L645-646: is the author suggesting that Zimmermann’s telome theory is an example of Fuzzy Arberian Morphology? I have always understood Zimmermann’s approach to be quite typological. He does talk about the gradual evolution of leaves, but he envisions this taking place by the rather rigid transformation of one structural category into another. I can see that Zimmerman does speak about the processes that are involved in these transformations, but his processes are purely hypothetical (and also are categorical). The Zimmermann processes have never been shown, to my knowledge, to actually exist in nature. They seem to me to be ideal transformations of one typological form into another. I do not understand how Zimmerman’s work is a good example of process morphology.

It is evident that also developmental geneticists started to take over some of Zimmermann’s ideas on land plant evolution. Of course, these hypothetical transformations in the sense of Zimmermann’s “telome theory” need to be understood in terms of the genetic regulation of plant developmental processes, as discussed by Floyd & Bowman (2007 in Int. J. Plant Sci. ), Harrison & al. (2005 in Nature), Harrison & Morris (2018 in Philos.Trans.R. Soc. B Biol. Sci.). Molecular data as presented by Beerling & Fleming (2007, Curr. Opin. Plant Biol.) in their paper “Zimmermann's telome theory of megaphyll leaf evolution” demonstrated mechanisms that explain the processes of overtopping and planation. However, only limited evidence is available for the webbing process that can possibly be substituted by lateral outgrowth during the formation of the blade (see also Melo-de-Pinna & Cruz 2020 in the book “Plant Ontogeny” (Demarco D. ed).

L674-675: I do not understand this sentence. “From a developmental perspective, canonical plant shoots (as generally recognized in vascular plants) can be defined as a process, i.e., they develop iteratively from an apex to produce lateral organs.” I do not understand what it means for a shoot to develop iteratively. Also, are shoots not process combinations? I do not understand how a shoot can be equated with a single process.

The reviewer critizes the process-oriented review on EVO-DEVO in ferns by Plackett & al. (2015, in Front. Plant Sci.)! Shoot apices in ferns and other vascular plants show repeated (= iterated) initiation of leaves (= lateral organs).

L675-676: The protonema of a moss does not have an axial body plan. The above ground stems of the mature gametophyte do. I do not think that there is any controversy about the fact that the mature gametophytes are at least shoot-like. I am not sure why the author thinks this is an important point.

L671-681: I do not understand what the author wants me to know after reading this paragraph. He begins by talking about process morphology and ends by talking about the structure of river-weeds shoots, but he does not relate these structures to process morphology.

Lines 676 – 682 contain sentences on process homology (i.e., homology between developmental processes) complementing structural homology. Ref1 should be aware that Jaramillo and Kramer (2007 in Int. J. Plant Sci., p.69) wrote: “Process homology reects the common inheritance of developmental genetic pathways or modules that can be co-opted to function in diverse situations. For this reason, process homology is dissociable from structural homology”, allowing the acceptance of “partial homology” sensu Rolf Sattler (1984, in Syst.Bot.; 1994 in Hall’s book on “Homology”), as well as the evolution of novel plant organs. Process thinking allows to accept completely flattened creeping plant bodies (with dorsiventral symmetry and apical growth) as something like highly modified shoots (e.g. in Marchantia-like liverwords and certain river-weeds (Podostemaceae).

L690-704: I think that the author makes some good points in the first part of this paragraph, but I feel that these points are diluted by the long quote at the end of the paragraph. The two parts of the paragraph do not seem closely related to me.

There are no special comments by the referee to the lines 706 – 863; but see general comments above…

Reviewer 2 Report

In this article the author examined several viewpoints in the Plant Morphology in the past, as a representative field of botany. Most researchers are, unconsciously, used to view plant organs in a clear-cut manner. But some researchers have felt the incompleteness of such view in understanding real plant morphogenesis. These researchers have struggled to establish alternative views that is represented by process-based understandings or the Fuzzy Arberian Morphology. As discussed here, deep understandings on a few species of ‘Model’ plants enabled us to examine ‘unusual’ or ‘fuzzy’ morphology in non-model plants nowadays, and developmental-genetics-based approaches have revealed the utility of the process-based understandings. But such ‘new-comers’ of non-model research often lack knowledge on the past debates on the above points. In this meaning, this review is quite timely and useful and educational to many young scientists in the plant development and plant morphology research fields. In addition, this review covers wide range of interesting publications from the past and to the latest, thus this is quite useful for education of students.

So I enjoyed to read this, but in honest, I felt that this is a little bit too long and wordy. Some redundancy might be possible to be fixed. And considering the presence of readers who are unfamiliar to the past debates between the clear-cut view and the Arberina view, it might be better to move the part 3.8 which examined the PSO terminology to the earlier part. Because this part strongly proved the vagueness and weakness of the clear-cut view in the plant morphology.

I have some additional comments as follows, that may be useful to increase the impact of this lovely essay. Some are minor and I do not insist on them.

  • The author briefly mentioned to epiphyllous buds here. In relation to it, the report on Kalanchoe by Garces et al. (2007) PNAS 104: 15578-155 might be effectively cited.
  • The term ‘higher plants’ (Line 400) is inappropriate, because in the past this word was set with ‘lower plants’ which included fungi and brown algae that are nowadays understood to be not involved in the ‘plant’ lineage.
  • Italicized ‘Arabidopsis’ (e.g., Line 415) is better to be roman-styled ‘Arabidopsis’ as many recent publications do. This is because in a strict sense, italicized Arabidopsis means the genus Arabidopsis. Moreover, the full spelling of the latin name of thale cress appeared much later in this article (Line 1187). This is better to be moved in the first appearance in the text.
  • The term ‘returned’ in Line 680 seems to be inappropriate because Marchantia-type body plan was not proven to be an ancestral than ‘leafy’ body plan, based on recent studies of land-plant molecular lineage.
  • The discussion in Lines 809-814 is a little bit delicate, because KNOX1 is ‘correlated with’ leaflet formation as written here, but is NOT proven to ‘identify’ it.

Author Response

In this article the author examined several viewpoints in the Plant Morphology in the past, as a representative field of botany. Most researchers are, unconsciously, used to view plant organs in a clear-cut manner. But some researchers have felt the incompleteness of such view in understanding real plant morphogenesis. These researchers have struggled to establish alternative views that is represented by process-based understandings or the Fuzzy Arberian Morphology. As discussed here, deep understandings on a few species of ‘Model’ plants enabled us to examine ‘unusual’ or ‘fuzzy’ morphology in non-model plants nowadays, and developmental-genetics-based approaches have revealed the utility of the process-based understandings. But such ‘new-comers’ of non-model research often lack knowledge on the past debates on the above points. In this meaning, this review is quite timely and useful and educational to many young scientists in the plant development and plant morphology research fields. In addition, this review covers wide range of interesting publications from the past and to the latest, thus this is quite useful for education of students.

My comment: I am happy that the second referee (“Ref2”) added these sentences. Some of them I took over in my introduction of the revised manuscript version.  

So I enjoyed to read this, but in honest, I felt that this is a little bit too long and wordy. Some redundancy might be possible to be fixed.

I tried to shorten some paragraphs…

And considering the presence of readers who are unfamiliar to the past debates between the clear-cut view and the Arberian view, it might be better to move the part 3.8 which examined the PSO terminology to the earlier part. Because this part strongly proved the vagueness and weakness of the clear-cut view in the plant morphology.

I moved part 3.8 to 3.1.

I have some additional comments as follows, that may be useful to increase the impact of this lovely essay. Some are minor and I do not insist on them.

  • The author briefly mentioned to epiphyllous buds here. In relation to it, the report on Kalanchoe by Garces et al. (2007) PNAS 104: 15578-155 might be effectively cited.

I added Garces & al 2007 in the revised version of my ms.

  • The term ‘higher plants’ (Line 400) is inappropriate, because in the past this word was set with ‘lower plants’ which included fungi and brown algae that are nowadays understood to be not involved in the ‘plant’ lineage.

“Higher plants” replaced by “vascular plants”

  • Italicized ‘Arabidopsis’ (e.g., Line 415) is better to be roman-styled ‘Arabidopsis’ as many recent publications do. This is because in a strict sense, italicized Arabidopsis means the genus Arabidopsis. Moreover, the full spelling of the latin name of thale cress appeared much later in this article (Line 1187). This is better to be moved in the first appearance in the text.

Everywhere in the ms I replaced the italicized ‘Arabidopsis’ and ‘Antirhinum’ by the roman-styled ‘Arabidopsis’ and ‘Antirhinum’. English names for these model plants omitted at all (not necessary). 

  • The term ‘returned’ in Line 680 seems to be inappropriate because Marchantia-type body plan was not proven to be an ancestral than ‘leafy’ body plan, based on recent studies of land-plant molecular lineage.

Whole sentence omitted in revised manuscript.

  • The discussion in Lines 809-814 is a little bit delicate, because KNOX1 is ‘correlated with’ leaflet formation as written here, but is NOT proven to ‘identify’ it.

I agree!!!

Round 2

Reviewer 2 Report

I am 100% satisfied with the revision.